# Continuous endosomes form functional subdomains and orchestrate rapid membrane trafficking in trypanosomes

Fabian Link[1], Alyssa Borges[1], Oliver Karo[1], Marvin Jungblut[2], Thomas Müller[1], Elisabeth Meyer-Natus[1], Timothy Krüger[1], Stefan Sachs[2], Nicola G Jones[1], Mary Morphew[3], Markus Sauer[2], Christian Stigloher[4], J Richard McIntosh[3], Markus Engstler[1]*

[1]Department of Cell & Developmental Biology, Biocentre, University of Würzburg, Würzburg, Germany; [2]Department of Biotechnology & Biophysics, Biocentre, University of Würzburg, Würzburg, Germany; [3]Molecular, Cellular & Developmental Biology, University of Colorado Boulder, Boulder, United States; [4]Imaging Core Facility, Biocentre, University of Würzburg, Würzburg, Germany

*For correspondence:
markus.engstler@biozentrum.
uni-wuerzburg.de

Competing interest: The authors declare that no competing interests exist.

**Abstract** Endocytosis is a common process observed in most eukaryotic cells, although its complexity varies among different organisms. In *Trypanosoma brucei*, the endocytic machinery is under special selective pressure because rapid membrane recycling is essential for immune evasion. This unicellular parasite effectively removes host antibodies from its cell surface through hydrodynamic drag and fast endocytic internalization. The entire process of membrane recycling occurs exclusively through the flagellar pocket, an extracellular organelle situated at the posterior pole of the spindle-shaped cell. The high-speed dynamics of membrane flux in trypanosomes do not seem compatible with the conventional concept of distinct compartments for early endosomes (EE), late endosomes (LE), and recycling endosomes (RE). To investigate the underlying structural basis for the remarkably fast membrane traffic in trypanosomes, we employed advanced techniques in light and electron microscopy to examine the three-dimensional architecture of the endosomal system. Our findings reveal that the endosomal system in trypanosomes exhibits a remarkably intricate structure. Instead of being compartmentalized, it constitutes a continuous membrane system, with specific functions of the endosome segregated into membrane subdomains enriched with classical markers for EE, LE, and RE. These membrane subdomains can partly overlap or are interspersed with areas that are negative for endosomal markers. This continuous endosome allows fast membrane flux by facilitated diffusion that is not slowed by multiple fission and fusion events.

## eLife assessment

This **important** study combines a range of advanced ultrastructural imaging approaches to define the unusual endosomal system of African trypanosomes. **Compelling** images reveal that, unlike a conventional set of compartments, the endosome in these protists forms a continuous membrane system with functionally distinct subdomains, as defined by canonical markers for early, late, and recycling endosomes. The findings compellingly support that the endocytic system in bloodstream stages has adapted to support remarkably high rates of membrane turnover necessary for immune complex removal and survival in the blood. This research is particularly relevant to those investigating infectious diseases

## Introduction

Endocytosis research has traditionally focussed on a limited number of organisms, primarily opisthokonts (including yeast and humans). Recently, using the extensive genetic and cell biological resources now available for studying numerous other organisms, studies on protists such as *Dictyostelium*, *Giardia*, *Toxoplasma*, or trypanosomes have revealed remarkable deviations from the hitherto existing understanding of the processes of endo- and exocytosis described in textbooks (reviewed in *Benchimol and de Souza, 2022*; *Link et al., 2021*; *McGovern et al., 2021*; *Vines and King, 2019*). Looking at the distinct evolutionary solutions for endosomal recycling across different branches of the eukaryotes, these protists, and especially the parasites among them, are of particular interest.

Throughout their life cycle, parasites often encounter a series of inherently hostile environments. The arms race between the parasite and its host has resulted in coevolutionary processes characterized by intense reciprocal selective pressures. Although parasite and host use similar basic cell biological pathways, parasites have often optimized these, either to achieve maximum efficiency or to establish additional functions. In this regard, the endocytosis and membrane recycling mechanisms observed in African trypanosomes serve as a remarkable illustration of the exceptional functional adaptations exhibited by parasites in response to their specific host environments.

*Trypanosoma brucei* is a representative of the group Discoba, which diverged before the appearance of opisthokonts (*Burki et al., 2020*; *Strassert et al., 2021*). *T. brucei* is an exclusively extracellular parasite, that resists the harsh conditions within the bloodstream, interstitial fluids, and lymph and thus thrives in the body fluids of its vertebrate host. Additionally, the cells have adapted to entirely distinct, yet equally challenging environments within the transmitting insect vector, *Glossina* (tsetse fly). Within the midgut of the fly, the trypanosomes proliferate to very high densities and, following a several week long journey, attach to the insect salivary glands before being injected back into the host skin during the bloodmeal of the tsetse (*Rose et al., 2020*).

The extracellular lifestyle of *T. brucei* renders the parasite vulnerable to constant attacks by the host's immune system. In response, these parasites have evolved a protective surface coat covering the entire plasma membrane (*Cross, 1975*). This coat comprises a single type of variant surface glycoprotein (VSG) and forms the basis of antigenic variation (*Mugnier et al., 2016*). *T. brucei* possesses hundreds of VSG genes that encode structurally similar yet antigenically distinct proteins (*Taylor and Rudenko, 2006*). However, only one VSG gene is expressed at a time, originating from a polycistronic expression site located at the telomere of the chromosomes (*Becker et al., 2004*; *Chaves et al., 1999*). The stochastic switching to the expression of a new VSG guarantees that the parasites remain ahead of the host immune response.

Antigenic variation is not the sole defense strategy employed against the host's immune reaction. During immune attack, parasites actively remove anti-VSG antibodies from their cell surface. This process of antibody clearance is driven by hydrodynamic drag forces resulting from the continuous directional movement of trypanosomes (*Engstler et al., 2007*). The VSG–antibody complexes on the cell surface are dragged against the swimming direction of the parasite and accumulate at the posterior pole of the cell. This region harbours an invagination in the plasma membrane known as the flagellar pocket (FP) (*Gull, 2003*; *Overath et al., 1997*). The FP, which marks the origin of the single attached flagellum, is the exclusive site for endo- and exocytosis in trypanosomes (*Gull, 2003*; *Overath et al., 1997*). Consequently, the accumulation of VSG–antibody complexes occurs precisely in the area of bulk membrane uptake. For the process of antibody clearance to function efficiently, several conditions must be met. The VSG coat must be densely packed to prevent immune recognition of non-variant surface proteins, VSG must be highly mobile to allow the drag forces to take effect, and endocytosis must be sufficiently rapid to internalize the VSG–antibody complexes effectively. Furthermore, endosomal sorting and membrane recycling must keep pace with the bulk membrane uptake to maintain membrane balance. In fact, the VSG coat is as densely packed as physically possible (*Hartel et al., 2016*) and VSG molecules possess a flexible C-terminal domain attached via a glycosylphosphatidylinositol (GPI) anchor to the outer leaflet of the plasma membrane (reviewed in *Borges et al., 2021*). This lipid anchoring enables the highly dynamic behaviour of the VSG coat (*Bartossek et al., 2017*), further facilitated by N-glycans that insulate VSG molecules, thereby minimizing protein–protein interactions (*Hartel et al., 2016*). VSG molecules exhibit diffusion on the plasma membrane with coefficients that enable fast randomization (*Hartel et al., 2016*; *Schwebs et al., 2022*). This rapid VSG diffusion is precisely balanced with membrane recycling kinetics, with

one surface equivalent being internalized and recycled within 12 min (*Engstler et al., 2004*). This exceptionally fast endocytosis is crucial for antibody clearance.

To summarize, *T. brucei* has evolved a series of interconnected cell biological adaptations in order to evade the immune system, ranging from a highly polarized cell structure and constant directional motion to the generation of a variable, densely packed yet highly mobile cell surface coat, as well as an unusually rapid endocytosis machinery. The strong selective pressure acting on trypanosomes has led to the integration and coordination of seemingly unrelated features, such as cell motility, surface protein dynamics, clathrin-mediated endocytosis, protein sorting, and membrane recycling. This illustrates the extraordinary cell biology of parasitic protozoa, which offers functional linkages beyond that of yeast or mammalian cells.

In *T. brucei*, the process of endocytosis (reviewed in *Link et al., 2021*) begins with the formation of clathrin-coated vesicles (CCVs) at the FP membrane (*Allen et al., 2003*; *Morgan et al., 2001*). Subsequently, both membrane-bound VSG and fluid-phase cargo pass through early endosomes (EE) (*Engstler et al., 2004*; *Grünfelder et al., 2003*). From this compartment, both either move directly to the recycling endosomes (RE) and return to the cell surface or first pass from the EE to the late endosomes (LE) before returning to the surface via the RE (*Engstler et al., 2004*). Alternatively, fluid-phase cargo can be directed from LEs to the lysosome for degradation (*Engstler et al., 2004*). In comparison, less is known about the biosynthetic pathway to the lysosome. Endogenous proteins like TbCatL, a soluble thiol protease, and p67, a membrane glycoprotein, are segregated from secretory cargo in the Golgi apparatus. They are then transported out of the Golgi, probably in a clathrin-dependent manner (*Tazeh et al., 2009*). An alternative route to the lysosome was discovered by disrupting transport signals in endogenous proteins, for example by blocking of GPI anchor addition to VSGs (*Triggs and Bangs, 2003*).

Similar to other eukaryotic organisms, the trafficking process through various endosomal compartments is regulated by a core set of three small Rab-GTPases (*Ackers et al., 2005*). TbRab5 and TbRab11 are responsible for controlling trafficking through the EE and RE, respectively (*Field et al., 1998*; *Hall et al., 2005*; *Pal et al., 2003*). TbRab7 plays a role in regulating late endocytic trafficking to the lysosome, but does not appear to be involved in the biosynthetic trafficking of either native lysosomal proteins or default reporters (*Silverman et al., 2011*). TbRab5A-positive endosomal structures have been described by immuno-electron microscopy as circular or elongated cisternae, while TbRab7-positive structures exhibit an irregular shape (*Engstler et al., 2004*; *Grünfelder et al., 2003*; *Overath and Engstler, 2004*). Additionally, TbRab11-positive structures are characterized as either elongated cisternae or disk-shaped exocytic carriers (EXCs) (*Grünfelder et al., 2003*). Apart from EXCs, the exocyst is currently the only proposed mediator of exocytosis (*Boehm et al., 2017*).

A thorough examination of the kinetics of endosomal recycling has validated the rapid passage through various endosomal compartments (*Engstler et al., 2004*; *Grünfelder et al., 2003*). However, these analyses have not addressed how the kinetics can be reconciled with the occurrence of multiple fission and fusion events that are thought to accompany endosomal transport through these compartments.

Here, we have employed a comprehensive set of techniques to scrutinize the ultrastructure of specific endosomal subdomains in *T. brucei*. In addition to utilizing electron tomography, super-resolution fluorescence microscopy, and colocalization analysis, we have introduced Tokuyasu sectioning for 3D immuno-electron microscopy. These tools allowed us to elucidate the structural basis for the remarkable speed of endosomal recycling in trypanosomes.

## Results

### The endosomal apparatus of *T. brucei* is a large intricate structure

In order to generate an overview of the endosomal ultrastructure in bloodstream form (BSF) trypanosomes, we performed electron tomography after fluid-phase cargo uptake assays with horseradish peroxidase (HRP) and ferritin (*Figure 1* and *Video 1*). Fluid-phase cargo was selected as it ensures a comprehensive labelling of all endosomal compartments (*Engstler et al., 2004*). Following HRP/ferritin endocytosis, the cells were fixed and embedded in Epon resin. To visualize HRP, photooxidation with 3,3′-diaminobenzidine (DAB) was employed before acquiring tomographic data (*Figure 1A, B, D*). Analysis of the reconstructed electron tomograms of various BSF cells revealed a morphologically

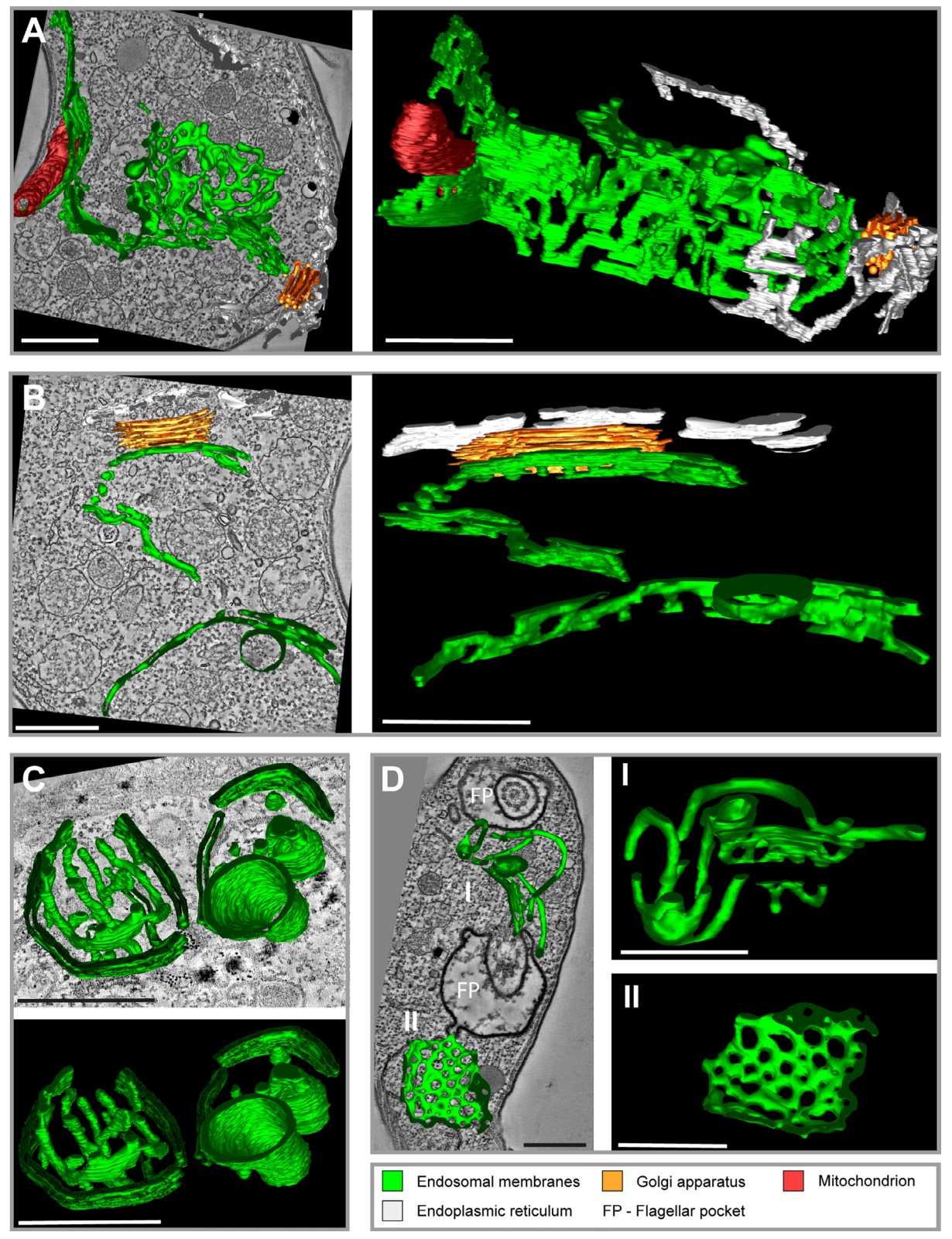

**Figure 1.** Tomographic reconstructions of the endosomal apparatus of *T. brucei* after cargo uptake reveal a large and mostly continuous membrane system. (**A**) Elongated and highly fenestrated endosomal sheets and palisades. (**B**) Elongated and slightly fenestrated endosomes. (**C**) Large circular endosomal cisternae. (**D**) Two different substructures are displayed: one contains tubular palisades (**I**) while the other one is a heavily fenestrated sheet (**II**). Horseradish peroxidase (HRP) endocytosis and 3,3'-diaminobenzidine (DAB) photooxidation were performed prior to tomogram acquisition (**A,**

*Figure 1 continued on next page*

*Figure 1 continued*

**B, D**). Ferritin endocytosis was performed prior to tomogram acquisition (**C**). All sections are 250 nm thick. The reconstructions (**A**) and (**D**) are based on three and two sections, respectively. The reconstructions (**B**) and (**C**) are based on one section each. The section and corresponding tomogram (**C**) originate from the same sample block as the image in Figure 4I from *Engstler et al., 2004*. Endosomal membranes are shown in green. The endoplasmic reticulum is visualized in white colour. The Golgi apparatus is labelled in orange and the mitochondrion is shown in red. Abbreviation: flagellar pocket (FP). Scale bars: 500 nm. Movies corresponding to panel A–D can be found here: (**A**) *Figure 1—video 1*, *Figure 1—video 2*; (**B**) *Figure 1—video 3*, *Figure 1—video 4*; (**C**) *Figure 1—video 5*, *Figure 1—video 6*; (**D**) *Figure 1—video 7*, *Figure 1—video 8*. The tomograms provided serve as a representative sample among the 37 tomograms that were recorded.

The online version of this article includes the following video(s) for figure 1:

**Figure 1—video 1.** z-stack of electron tomogram from *Figure 1A*.
https://elifesciences.org/articles/91194/figures#fig1video1

**Figure 1—video 2.** 3D model of electron tomogram from *Figure 1A*.
https://elifesciences.org/articles/91194/figures#fig1video2

**Figure 1—video 3.** z-stack of electron tomogram from *Figure 1B*.
https://elifesciences.org/articles/91194/figures#fig1video3

**Figure 1—video 4.** 3D model of electron tomogram from *Figure 1B*.
https://elifesciences.org/articles/91194/figures#fig1video4

**Figure 1—video 5.** z-stack of electron tomogram from *Figure 1C*.
https://elifesciences.org/articles/91194/figures#fig1video5

**Figure 1—video 6.** 3D model of electron tomogram from *Figure 1C*.
https://elifesciences.org/articles/91194/figures#fig1video6

**Figure 1—video 7.** z-stack of electron tomogram from *Figure 1D*.
https://elifesciences.org/articles/91194/figures#fig1video7

**Figure 1—video 8.** 3D model of electron tomogram from *Figure 1D*.
https://elifesciences.org/articles/91194/figures#fig1video8

intricate system of endosomes comprised of interconnected and fenestrated membrane networks. The main endosome, spanning the posterior region of the cell from the vicinity of the FP to the Golgi apparatus, exhibited elongated (*Figure 1A, B*) and circular membrane sheets (*Figure 1C, D*), encompassing a volume with a diameter of approximately 2 µm and a length of 4 µm. The localization of endosomes was strictly confined to the posterior cytoplasm. Their precise positioning, however, appeared to be flexible, indicating a lack of direct and rigid connection with the microtubule cytoskeleton. This observation is not surprising, as there are no reports of cytoplasmic microtubules in trypanosomes. All microtubules appear to be either subpellicular or within the flagellum. However, it is very likely that a scaffold is necessary to maintain the intricate membrane structure of the endosomal system, especially given the constant deformation of the cell body by the beating of the attached flagellum. The electron tomograms further suggest that the large size of the endosomes may render them unable to pass through the space between the nucleus and pellicular microtubules. Another notable finding was the proximity of endosomes to the Golgi apparatus (*Figure 1A, B*), suggesting a potential involvement of the endosomal system in post-Golgi transport.

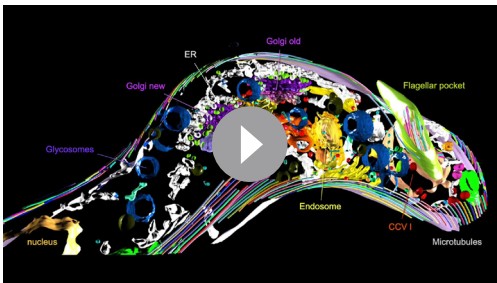

**Video 1.** Tomographic reconstructions of *T. brucei*. The movie is available online and includes annotations for better understanding.
https://elifesciences.org/articles/91194/figures#video1

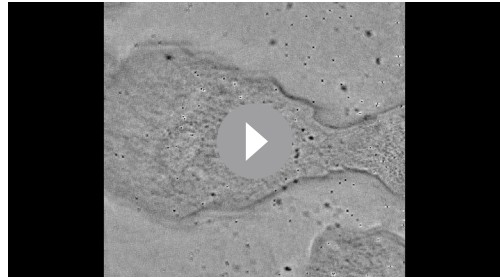

**Video 2.** Tomographic reconstructions of *T. brucei*. Cells were chemically fixed before further processing. Scale bar: 500 nm.
https://elifesciences.org/articles/91194/figures#video2

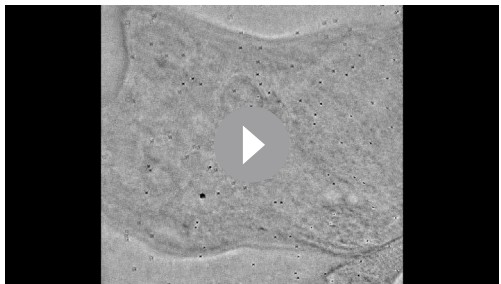

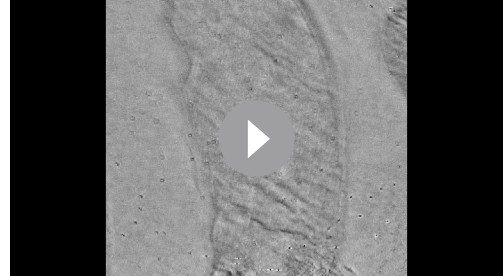

**Video 3.** Tomographic reconstructions of *T. brucei*. Cells were chemically fixed before further processing. Scale bar: 500 nm.

https://elifesciences.org/articles/91194/figures#video3

**Video 4.** Tomographic reconstructions of *T. brucei*. Cells were chemically fixed before further processing. Scale bar: 500 nm.

https://elifesciences.org/articles/91194/figures#video4

To investigate potential negative impacts of processing steps (cargo uptake, centrifugation, washing) on endosomal organization, we directly fixed cells in the cell culture flask, embedded them in Epon, and conducted electron tomography. The resulting tomograms revealed endosomal organization consistent with that observed in cells fixed after processing (see *Video 2*, *Video 3*, and *Video 4*).

In summary, electron tomography analyses provided evidence that the endosomal system in bloodstream form (BSF) cells is a complex, flexible, and largely continuous membrane network.

## Super-resolution microscopy reveals the endosomal apparatus as a largely continuous structure

To be able to do specific labelling of the endosomal compartment, we first tested the potential of various super-resolution light microscopy techniques to resolve and corroborate the structure of the entire endosomal compartment (*Figure 2*, *Figure 2—figure supplement 1*, *Figure 2—figure supplement 2*). To allow the identification of endosomes, we utilized the marker protein EP1 that had previously been demonstrated to label the entire endosome and FP of BSF trypanosomes (*Engstler and Boshart, 2004*). EP1, a cell surface protein found in insect stage trypanosomes, is mostly excluded from the VSG coat of (BSF) trypanosomes.

To utilize EP1 as an endosomal membrane marker in *direct* stochastic optical reconstruction microscopy (*d*STORM) (*Heilemann et al., 2008*), an EP1::HaloTag expressing cell line was generated (*Figure 2—figure supplement 2*). The transgenic parasites were then labelled with a HaloTag ligand conjugated to Alexa Fluor 647 (custom conjugated, see Methods), and imaged in 100 mM ethanolamine (MEA) photo-switching buffer (*Figure 2—figure supplement 1*).

As expected, the EP1::HaloTag signal was localized in the posterior region of the cell, between the nucleus and kinetoplast (*Figure 2—figure supplement 1*, I, II). Although the entire endosomal compartment seemed to be labelled and gave the appearance of a largely connected system (*Figure 2—figure supplement 1*, III, IV), the labelling density was not consistently strong, resulting in occasionally thin membrane bridges (*Figure 2—figure supplement 1*, upper row, image IV, indicated by an arrow). In addition, the 3D volume of the labelled region was too large (approx. 4 × 4 × 5 µm) to be completely captured by single-molecule localization microscopy. On top of that, not all cells exhibited an EP1::Halo signal, indicating heterogeneous gene expression within the population (*Figure 2—figure supplement 1*, lower row, lower cell).

To overcome the non-uniform labelling density based on an endogenous marker protein and to be able to capture the entire 3D volume of the parasite, a combination of dextran uptake assays and expansion microscopy (ExM) (*Chen et al., 2015*) was performed to visualize the endosomes. The fluorescently labelled dextran was observed in the posterior region between the nucleus and kinetoplast in all cells (*Figure 2A*). The expanded and unexpanded cells revealed elongated cable-like structures representing the endosomes, spanning from the FP to the vicinity of the nucleus. The expansion process improved the resolution of the structure, allowing for the visualization of potential circular cisternae (*Figure 2A*, middle row, image III, arrow). While the FP labelling was visible in nearly all unexpanded cells, not all expanded cells exhibited a corresponding signal (*Figure 2A*). To improve

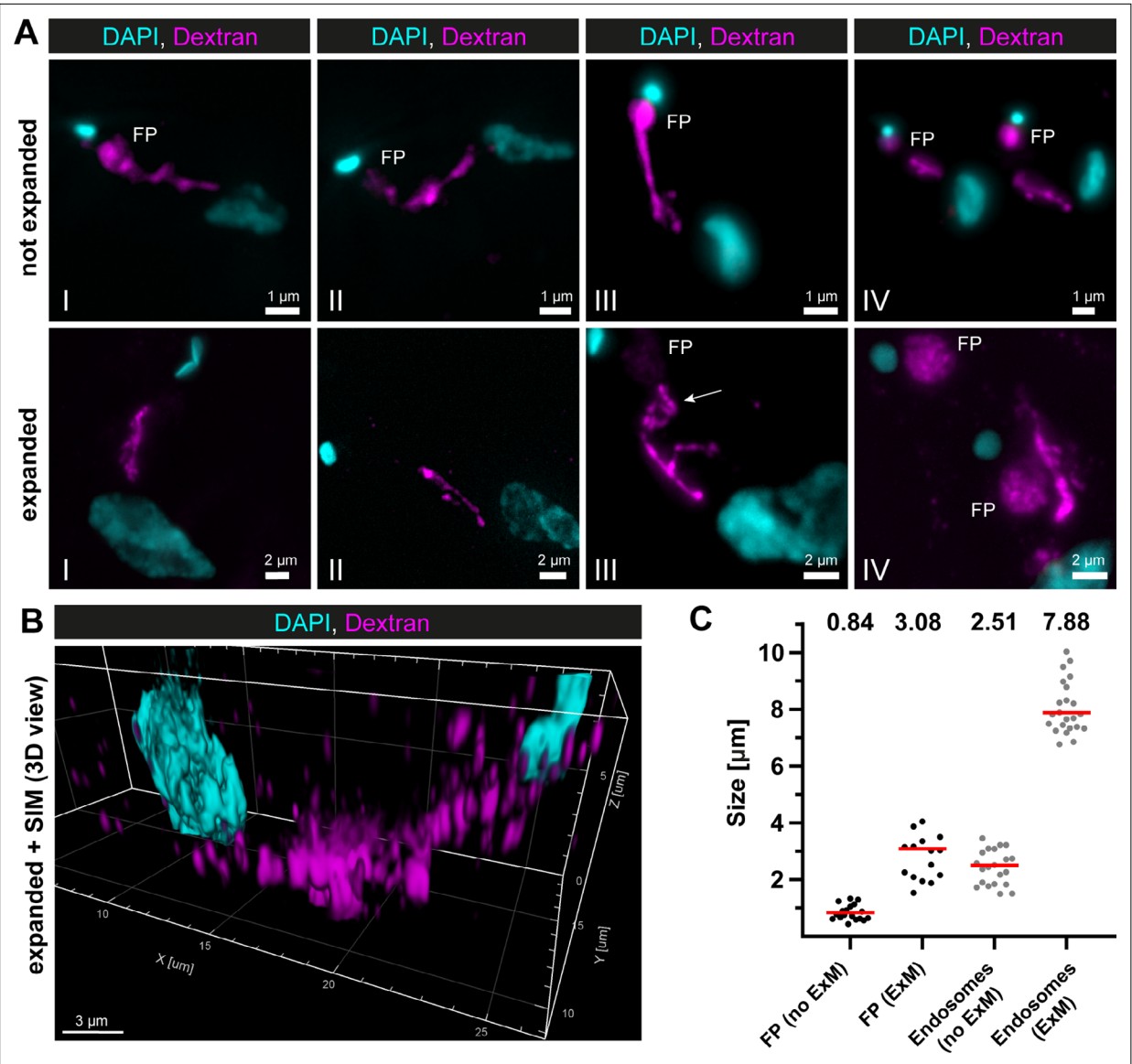

**Figure 2.** Super-resolution light microscopy reveals continuous endosomal membranes. (**A**) Visualization of the endosomal system in *T. brucei* using expansion microscopy (ExM). Bloodstream form cells were pulsed with dextran conjugated to Alexa488 and fixed with 4% formaldehyde and 0.2% glutaraldehyde. Shown are exemplary images (I–IV) of not expanded and expanded cells imaged using widefield microscopy. All images represent a merge of 4',6-diamidino-2-phenylindole (DAPI) staining (cyan) and dextran signal (magenta). Flagellar pockets (FP) are annotated. (**B**) 3D presentation of the endosomal system using a combination of ExM and structured illumination microscopy (SIM). The same gels as imaged in (**B**) were analysed with SIM. The 3D presentation was generated using the IMARIS package (Bitplane). (**C**) Scatter plot of the quantification of the expansion factor. The diameter of the fluorescence signal corresponding to FP and length of the endosomes were measured in multiple not expanded (no ExM) and expanded cells (ExM) ($n \geq 15$). The median values are indicated as red bars and the corresponding numbers are shown in the upper part of the panel.

The online version of this article includes the following source data and figure supplement(s) for figure 2:

**Figure supplement 1.** Visualization of the endosomal system in *T. brucei* by direct stochastic optical reconstruction microscopy (*d*STORM).

**Figure supplement 2.** Validation of the generated EP1::HaloTag cell line.

**Figure supplement 2—source data 1.** Integration PCR for generation of EP1::Halo cell line.

**Figure supplement 2—source data 2.** Integration PCR for generation of EP1::Halo cell line - labelled.

the resolution even further, we performed structured illumination microscopy (SIM) on expanded cells (*Figure 2B*). The result also demonstrated a largely continuous dextran signal between the nucleus and the kinetoplast, indicating the continuity of the endosomal structures.

To determine the expansion factor, the diameter of the FP and the length of the endosomes were measured in multiple expanded and unexpanded cells (*Figure 2C*). The median diameter of the fluorescence signal corresponding to the FP increased from 0.84 to 3.08 μm, resulting in an expansion factor of 3.6. The median length of the endosomes increased from 2.51 to 7.88 μm, yielding an expansion factor of 3.1.

In summary, super-resolution microscopy using a transgenic marker cell line and cargo uptake assays supported the possibility that in trypanosomes endosomes are largely continuous.

## Quantitative 3D colocalization analysis reveals overlaps between early, late, and recycling endosomal marker proteins

To functionally characterize different components of the endosomal system, specific antibodies against the endosomal marker proteins TbRab5A (EE), TbRab7 (late endosomes), and TbRab11 (RE) were generated and validated (*Figure 3—figure supplement 1*). Immunofluorescence assays were conducted, and the samples were imaged using 3D widefield microscopy. For quantitative colocalization analyses, the previously established endosomal marker EP1::GFP (*Engstler and Boshart, 2004*) was used to define the entire endosomal compartment as the region of interest (ROI) (*Figure 3*).

The EP1::GFP fluorescence revealed a variable number of high-intensity signals, surrounded, and connected by weaker signals (*Figure 3A*, I). The fluorescence of the TbRab marker proteins was present in foci of varying numbers, sizes, and intensities (*Figure 3A*, II– IV). The fluorescence signals for all three marker proteins were within the ROI defined by EP1. The visualization as maximum intensity projections (MIPs) suggested that the different subdomains exhibited a low degree of colocalization and could represent physically distinct compartments for EE, LE, and RE (*Figure 3A*, V). To quantitate this finding, 3D objects were generated from the fluorescent signals for colocalization analysis using IMARIS (*Figure 3A*, VI–X). These objects showed areas of unique staining as well as partially overlapping regions. The pairwise colocalization of different endosomal markers is shown in *Figure 3A*, XI–XIII and *Figure 3B*. The different cells in *Figure 3B* were chosen to represent the dynamic nature of the labelled structures. Consequently, the selected cells provide diverse examples of how the labelling can appear.

For the colocalization analysis, the EP1::GFP objects were masked and used as a 3D ROI in IMARIS. The colocalization matrix (*Figure 3C*) showed that TbRab5A and TbRab7 exhibited the highest signal overlap, with approximately 53% volume in both combinations. While 35% of the volume of TbRab5A was colocalized with TbRab11, 26.5% of the volume of TbRab11 was colocalized with TbRab5A. The differences in colocalization volumes can be attributed to the larger volume of the TbRab11-positive compartment. Similar observations were made for the combination of TbRab7 and TbRab11. While 45% of the volume of TbRab7 was colocalized with TbRab11, 29.9% of the volume of TbRab11 was colocalized with TbRab7. Furthermore, the different marker combinations yielded comparable moderate correlation coefficients (Pearson) ranging from 0.37 (TbRab5A and TbRab11) to 0.43 (TbRab5A and TbRab7) (*Figure 3D*).

In summary, the quantitative colocalization analyses revealed that on the one hand, the endosomal system features a high degree of connectivity, with considerable overlap of endosomal marker regions, and on the other hand, TbRab5A, TbRab7, and TbRab11 also demarcate separated regions in that system. These results can be interpreted as evidence of a continuous endosomal membrane system harbouring functional subdomains, with a limited amount of potentially separated EE, LE, or RE.

## The endosomal markers TbRab5A, TbRab7, and TbRab11 are present on the same membranes

To investigate the potential functional sub-compartmentalization of endosomes in trypanosomes, we conducted immunogold assays on Tokuyasu cryosections. We defined gold particles that were 30 nm or closer to a membrane as membrane-bound, considering a conservative calculation where two IgG molecules, each 15 nm in size, are positioned between the bound epitope and the gold particle (*Figure 4A*). Consequently, gold particles located further away may represent cytoplasmic TbRab

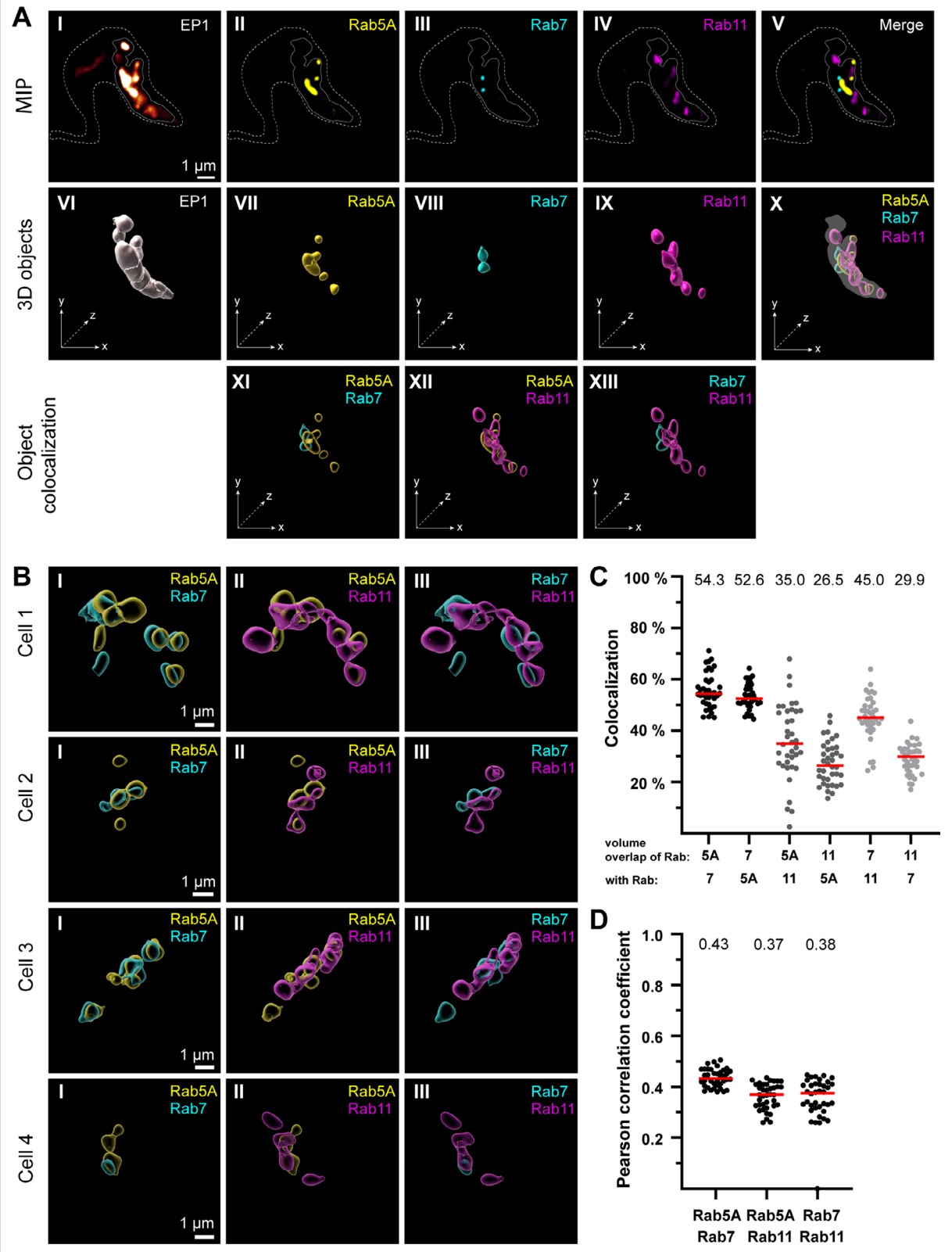

**Figure 3.** The endosomal markers TbRab5A, TbRab7, and TbRab11 partly colocalize in the posterior region of the cell. (**A**) The experimental workflow, from fluorescence images to object generation and quantitative 3D colocalization analysis, is exemplified. EP1::GFP (**I**) expressing cells were fixed and labelled with anti-TbRab5A (**II**), anti-TbRab7 (**III**), and anti-TbRab11 (**IV**) antibodies and imaged using widefield microscopy. Upper row: maximum intensity projections (MIPs) for each individual fluorescence channel (**I–V**) as well as a merge of all three anti-TbRab marker channels (**V**). The outline of

*Figure 3 continued on next page*

*Figure 3 continued*

the cell is shown by a dashed line (**I–V**) and the region of interest (ROI; endosomal system) is presented by a solid line (**I–V**). Middle row: corresponding 3D objects to display the ROI (**VI**) and the TbRab marker objects (**VII–IX**). The merge of all objects confirmed that the TbRab marker objects are located within the ROI (**X**). Lower row: colocalization of the objects representing TbRab5A and TbRab7 (**XI**), TbRab5A and TbRab11 (**XII**), and TbRab7 and TbRab11 (**XIII**). The *x*, *y*, and *z* axes are indicated to support the three-dimensional view. (**B**) Object colocalization shown for additional cells. The colocalization of TbRab5A and TbRab7 (**I**), TbRab5A and TbRab11 (**II**), and TbRab7 and TbRab11 (**III**) is shown for different cells (Cells 1–4). (**C, D**) Quantification of colocalization analysis for TbRab marker combinations. The colocalization function of IMARIS was used for the quantification. The EP1 surfaces defined the ROI (see panel A) and the automatic thresholding function (*Costes et al., 2004*) ensured minimal user bias. Each data point (*n* = 38) represents a field of view with 30–40 cells, corresponding to a total number of ~1200 analysed cells. The median is highlighted with a red line and the corresponding number is written on top of the dataset. (**C**) Scatter plot of the colocalization analysis for the different TbRab marker combinations using the percentage of volume overlap. (**D**) Scatter plot of the colocalization analysis for the different TbRab marker combinations using the Pearson correlation coefficient.

The online version of this article includes the following source data and figure supplement(s) for figure 3:

**Figure supplement 1.** Validation of different anti-TbRab antibodies.

**Figure supplement 1—source data 1.** Antibody validation western blots unlabelled.

**Figure supplement 1—source data 2.** Antibody validation western blots labelled.

proteins or, as the 'linkage error' can also occur in the *z*-plane, correspond to membranes that are not visible within the 55 nm thickness of the cryosection (*Figure 4C*, arrows). The immuno-labelled structures exhibited a range of architectural features, including vesicular and tubular cisternae with circular or elongated shapes (*Figures 4–6*). The tubular structures had a thickness of 20–30 nm and extended up to a few microns in length (with the longest example shown in *Figure 5F*, arrow). Vesicular shapes were observed in various sizes ranging from 30 to 135 nm in diameter.

While there were endosomal structures exclusively labelled with TbRab5A or TbRab7 (*Figure 4D, E, G*), a significant number of endosomal structures were labelled with both marker proteins (*Figure 4B–G*). In contrast to TbRab5A and TbRab7, TbRab11 tended to form signal clusters on elongated structures (*Figure 5A, B, D*; *Figure 6D–F*). TbRab11 was observed together with TbRab5A (*Figure 5A–F*) or TbRab7 (*Figure 6A–G*) on the same endosomal membranes, but it was also found as the sole marker on other endosomal membranes (*Figure 5B, D, F*; *Figure 6C, G*). The structures with a disk shape and positive for TbRab11 (*Figure 5F*) were identified as EXCs, as previously described in *Grünfelder et al., 2003*. Another noteworthy observation was that long elongated structures tended to have no or only weak labelling with any of the TbRab markers (*Figure 5F*, arrow). To exclude potential artefacts, both gold size combinations were utilized for all marker combinations.

In summary, the exemplary images shown in *Figures 4–6* demonstrate the colocalization of TbRab5A, TbRab7, and TbRab11 on the same membranes, suggesting that the endosomal membrane system is unlikely to consist of distinct and independent compartments for EE, LE, and RE.

To confirm that the analysed membrane structures indeed represent endosomal membranes, we conducted cargo uptake assays and anti-VSG immunogold labelling (*Figure 7*). Trypanosomes were either pulsed with fluid-phase cargo (bovine serum albumin [BSA]; *Figure 7A–C*) or cargo was taken up by receptor-mediated endocytosis (transferrin, Tf; *Figure 7D–F*), and the sections were subsequently labelled with anti-TbRab antibodies. BSA and transferrin perfused the entire endosomal system, with the conjugated 5 nm gold particles visible inside the endosomal structures (*Figure 7A–F*). Moreover, gold particles were observed within vesicles (*Figure 7B, D*) and accumulated within the lysosome (*Figure 7A, B*). In another set of experiments, the parasites were not pulsed with cargo markers, but the sections were labelled with anti-VSG and anti-TbRab antibodies (*Figure 7G–I*). The anti-VSG antibodies decorated the entire plasma membrane (*Figure 7G*) as well as intracellular membrane structures (*Figure 7G–I*). The different TbRab marker proteins were found on the same membrane structures as the cargo markers or the anti-VSG antibodies (*Figure 7*, all panels). In summary, the use of cargo markers and antibodies targeting VSG and endosomal marker proteins confirmed the identity of the trypanosome endosomes.

## 3D Tokuyasu demonstrates that trypanosomes have continuous endosomal membranes

Following the visualization of continuous endosomal membranes through cargo uptake assays in Epon resin sections (*Figure 1*), we aimed to confirm these findings using 3D immuno-electron microscopy on

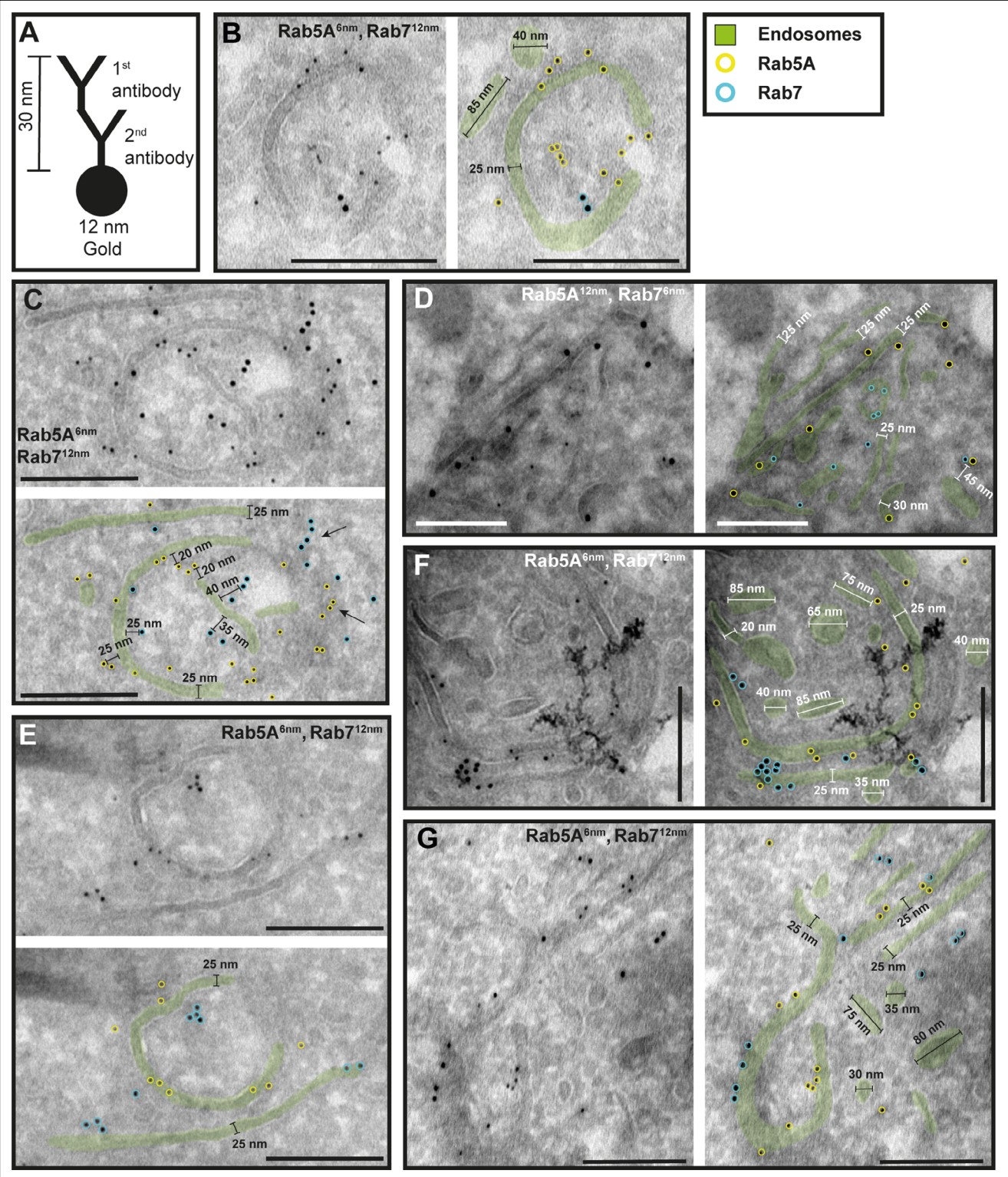

**Figure 4.** The endosomal markers TbRab5A and TbRab7 are present on the same membrane. (**A**) Schematic representation of antibody binding to visualize the maximum linkage error of 30 nm between a bound epitope and the gold particle. (**B–G**) Electron micrographs of cryosections labelled with rat anti-TbRab5A and guinea pig anti-TbRab7 antibodies and visualized with 6 or 12 nm gold-coupled secondary antibodies. For each panel, an annotated and an unedited version of the image is presented. Endosomes are highlighted in green. Gold particles corresponding to TbRab5A signals are labelled in yellow. Gold particles corresponding to TbRab7 signals are highlighted in cyan. Scale bars: 200 nm. The images shown are representative examples drawn from over 300 micrographs generated in four distinct labelling experiments.

*Figure 4 continued on next page*

*Figure 4 continued*

The online version of this article includes the following source data for figure 4:

**Source data 1.** Uncropped electron micrographs of *Figure 4*.

cryosections (*Figure 8*). Cryosections with a thickness of 250 nm were labelled with antibodies against TbRab5A, TbRab7, and TbRab11. The 3D reconstructions provided an in-depth view of the interconnected circular and tube-like structures. Although the tomograms represented only a 250-nm-thick section of a cell, it was evident that the membrane system likely exhibited further connections above and below the field of view. This assumption was confirmed by our experiments with conventional electron tomography. The 3D Tokuyasu method effectively highlighted the interconnectedness of these structures, expanding upon the observations from 2D analyses. Interestingly, these tomograms did not exhibit the fenestration pattern identified in traditional electron tomography. We suspect that this is due to methodological reasons. The Tokuyasu procedure uses only aldehydes to fix all structures. Consequently, effective fixation of lipids occurs only through their association with membrane proteins. Thus, the lack of visible fenestration is likely due to possible loss of lipids during incomplete fixation.

We observed endosomes decorated with different TbRab signals, supporting our assumption that functional subdomains exist within the same endosomal membrane system (*Figure 8A* II and III, B II). It is worth noting that smaller gold particles demonstrated better penetration of the cryosections, explaining the higher abundance of 6 nm gold particles compared to 12 nm gold particles. Another noteworthy finding was the presence of TbRab5A and TbRab7 in the lysosomal lumen, suggesting their possible degradation (*Figure 8A, B*).

In addition, the endosomal membranes were juxtaposed to the trans-Golgi in all Tokuyasu tomograms, further implying a role of the endosomal system in post-Golgi trafficking (*Figure 8A* I, B III). In summary, our second 3D electron microscopy approach successfully visualized the continuous membrane system of the endosomal apparatus in *T. brucei* and represents the first utilization of the advanced 3D Tokuyasu technique in trypanosomes.

## Discussion

Evolution has endowed African trypanosomes with a cellular multi-tool, the VSG-surface coat. Apart from exhibiting antigenic variation across hundreds of coat variants, the VSG proteins are attached to the cell surface through GPI moieties, enabling their remarkable mobility. Exploiting this feature, the parasites evade early immune responses by clearing antibodies. This evasion process hinges upon the highly polarized nature of the trypanosome cell body, with exclusive endocytic uptake occurring at the posterior end, coupled with an exceptionally rapid membrane recycling rate. The prevailing model of the endosomal system, consisting of distinct compartments for EE, LE, and RE, seems incompatible with the observed rapid turnover rates in *T. brucei*. Nevertheless, the parasite expresses the canonical Rab GTPases which serve as markers for these endosomal compartments, such as Rab5 (EE), Rab7 (LE), and Rab11 (RE). To resolve this apparent contradiction, we performed a structure–function analysis of the trypanosome endosome apparatus.

3D electron tomography reconstructions, utilizing cargo uptake assays (*Figure 1* and *Video 1* ), revealed a complex network of endosomal structures in *T. brucei*. These compartments were characterized by interconnected fenestrated sheets, circular cisternae, and tubular structures, that formed a single intricate endosome. Our results shine a new light on transmission electron microscopic (TEM) studies of the trypanosome endomembrane system (reviewed in *Link et al., 2021*). Based on 2D information the giant endosomal sheets had been interpreted as extended tubular structures, fenestrations as vesicles and the connective areas as irregularly shaped structures (*Engstler et al., 2004*; *Grünfelder et al., 2003*; *Overath and Engstler, 2004*).

In mammalian cells, tubular endosomes constitute a major part of the endosomal system. Tubules possess a higher surface area-to-volume ratio compared to vacuolar regions. As a result, they have been proposed to geometrically sort membrane-associated proteins over soluble cargo (*Mayor et al., 1993*). It is assumed that the carrier tubules, following detachment, transport their cargo either to the plasma membrane or to the RE.

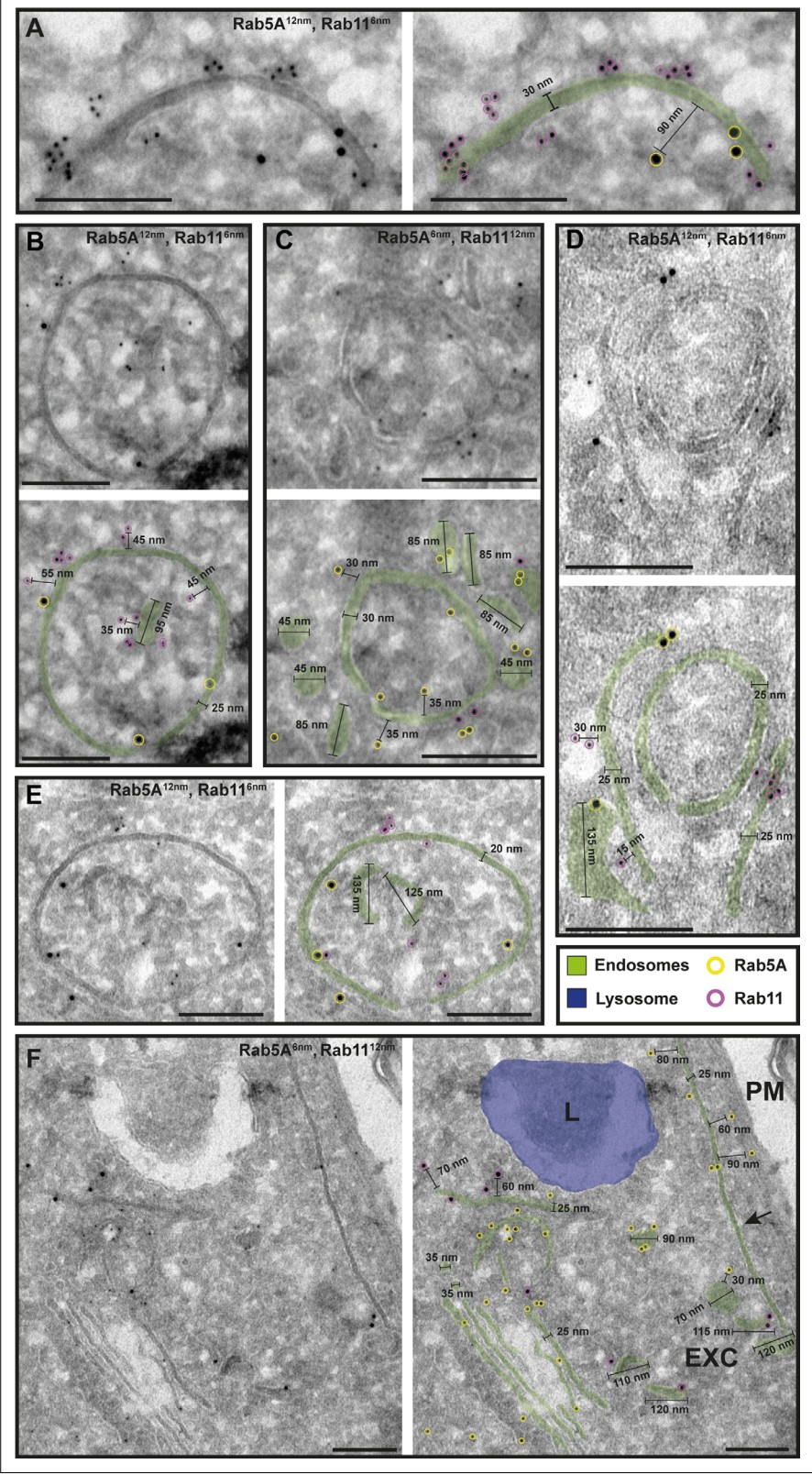

**Figure 5.** The endosomal markers TbRab5A and TbRab11 are present on the same membranes. (**A–F**) Electron micrographs of cryosections labelled with rat anti-TbRab5A and rabbit anti-TbRab11 antibodies and visualized with 6 or 12 nm gold-coupled secondary antibodies. For each panel, an annotated and an unedited version of the image is presented. Endosomes are marked in green, and the lysosome is highlighted in blue. Gold particles

*Figure 5 continued on next page*

*Figure 5 continued*

are marked in yellow (TbRab5A) and in magenta (TbRab11). Scale bars: 200 nm. Exocytic carrier (EXC), lysosome (L), and plasma membrane (PM). The displayed images are representative examples drawn from over 300 images generated in four distinct labelling experiments.

The online version of this article includes the following source data for figure 5:

**Source data 1.** Uncropped electron micrographs of *Figure 5*.

---

*T. brucei* lacks elongated tubulo-vesicular structures. Furthermore, our tomographic analysis did not reveal the conventional tubular carriers (reviewed in *Chi et al., 2015*). Instead, in these parasites, geometric sorting most likely occurs at the rims of flat, fenestrated sheets. A specific class of clathrin-coated vesicles (CCV II) pinches off from these strongly curved membrane areas. They transport bulky fluid-phase cargo and membrane protein complexes (such as VSG$^{IgG}$) to the lysosome (*Engstler et al., 2004*; *Grünfelder et al., 2003*).

The sorting mechanism into CCV II vesicles could be a purely physical process. Both the fenestrated sheets and circular cisternae exhibit the same inner width of about 35 nm. The VSG coat occupies 2 × 15 nm of this width, significantly limiting the available volume for fluid-phase cargo. As the VSG concentration in the membrane increases, the bulky cargo is pushed towards the edges of the sheets, resulting in further membrane deformation that facilitates the recruitment of clathrin. The VSG is excluded from the CCV II and thus not transported to the lysosome. How this works mechanistically is unknown, however, it is tempting to speculate that the geometric sorting of larger cargo into the CCV II, sterically excludes VSG.

The electron tomographic analysis of the *T. brucei* endosome yielded compelling evidence of an extensive and interconnected membrane system. While this conclusion is supported by a substantial number of tomograms, the technique is not well suited for quantitative analyses.

Therefore, we conducted super-resolution microscopy and cargo uptake assays (*Figure 2* and *Figure 2—figure supplement 1*). To analyse the full 3D volume of the cell, ExM was conducted (*Figure 2*). The visualized endosomal network looked like the elongated and circular structures found in the reconstruction of the electron tomograms (*Figure 1*). While the elongated structures could easily correspond to the fenestrated sheets (*Figure 2A*), the circular structure (*Figure 2A*, middle row, image III, arrow) nicely recapitulates the morphology of a circular cisternae (*Figure 1C*). Interestingly, dividing cells (possessing two kinetoplasts and two FPs) always showed a fluorescence signal corresponding to the endosomal system juxtaposed only to the 'old' FP, indicating that the duplication of the endosomal network might occur de novo during the cell cycle. This is currently speculation and requires more in-depth analysis. The 3D presentation (*Figure 2B*) confirmed the continuity of the endosomal membranes, which had been seen already in the electron tomography.

In contrast to *d*STORM or SIM, ExM offers a selective perspective on the trypanosome endosomal system. To date, ExM has been effectively employed to visualize cell structures associated with the cytoskeleton or other stabilizing supports (*Alonso, 2022*; *Amodeo et al., 2021*; *Gambarotto et al., 2019*; *Gorilak et al., 2021*; *Kalichava and Ochsenreiter, 2021*). However, we could not find any published ExM images of endosomes. Our initial efforts to identify endosomal markers using ExM were also unsuccessful. Only when we added the inert fluid-phase cargo dextran did the endosomes become clearly visible. This was likely due to the stabilizing effect of the cargo.

Having established that the endosomes in trypanosomes likely form a continuous cellular structure, our next inquiry was whether this would be evident in the simultaneous presence of the canonical marker proteins TbRab5A, TbRab7, and TbRab11. To define the ROI in 3D immunofluorescence microscopy, we utilized the previously established fluorescent marker EP1::GFP (*Engstler and Boshart, 2004*), which stains the entire endosomal system. MIPs of multichannel images indicated that the TbRabs were closely juxtaposed but in distinct endosomal compartments (*Figure 3A*, upper row). This finding aligns with the results of a previously published quantitative colocalization analysis of TbRab5A and TbRab7 (*Engstler et al., 2004*). However, the presentation as 3D objects already implied an overlap between the different marker proteins. The present study now reveals that endosomal sheets can easily span over several micrometres, which fundamentally alters the interpretation of the light microscopy colocalization data. Thus, the seemingly separate structures could well be subdomains of the same membrane structure.

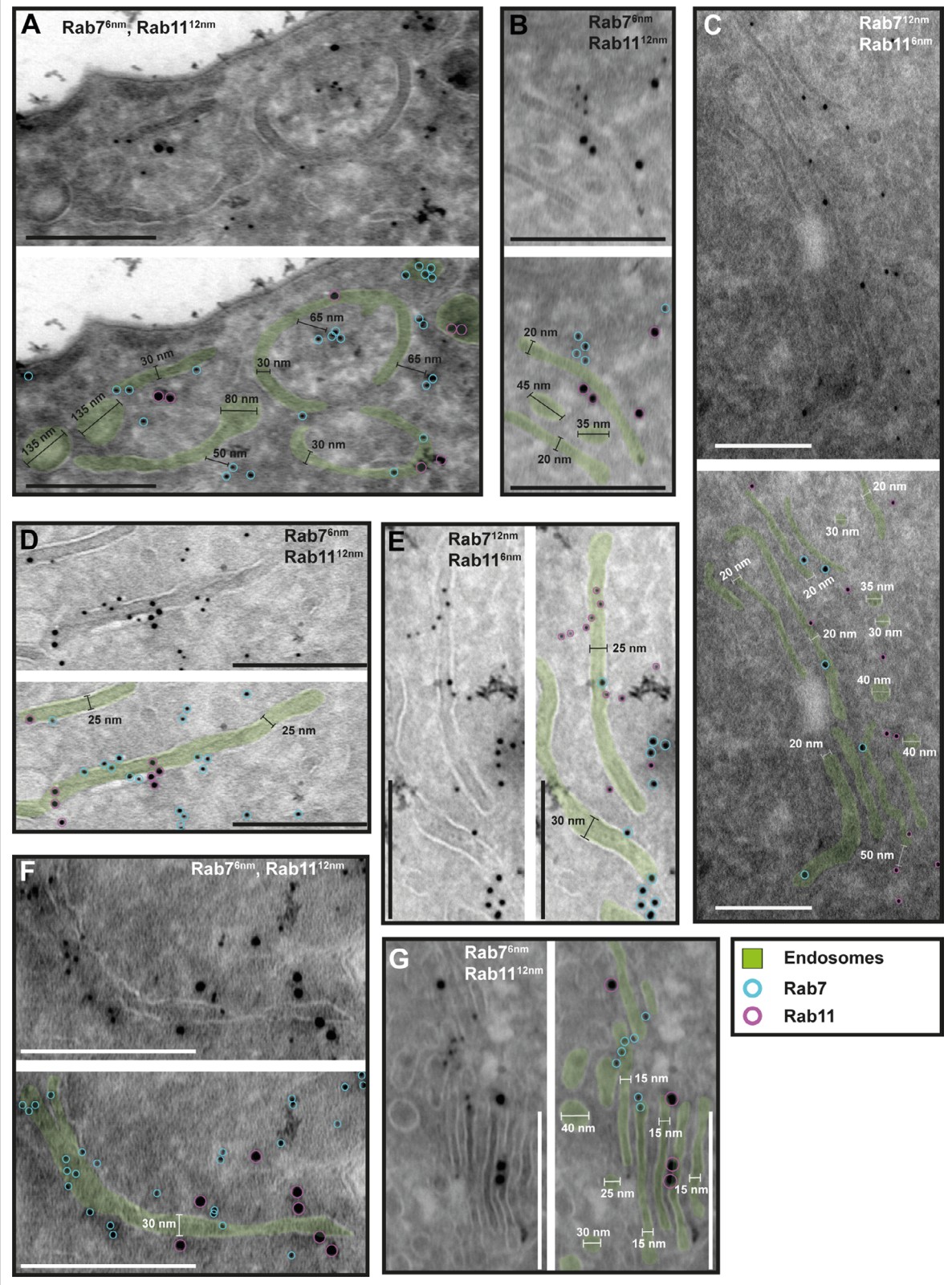

**Figure 6.** The endosomal markers TbRab7 and TbRab11 are present on the same membranes. (**A–G**) Electron micrographs of cryosections labelled with guinea pig anti-TbRab7 and rabbit anti-TbRab11 antibodies visualized with 6 or 12 nm gold-coupled secondary antibodies. For each panel, an annotated and an unedited version of the image is presented. Endosomes are highlighted in green. Gold particles are labelled in cyan (TbRab7) and

*Figure 6 continued on next page*

*Figure 6 continued*

in magenta (TbRab11). Scale bars: 200 nm. The displayed images are representative examples drawn from over 200 images generated in three distinct labelling experiments.

The online version of this article includes the following source data for figure 6:

**Source data 1.** Uncropped electron micrographs of *Figure 6*.

We therefore performed quantitative colocalization analyses with all marker combinations (*Figure 3*). The degree of colocalization was assessed by determining the percentage of overlap for the volume of one TbRab marker colocalized with another. Notably, TbRab5A and TbRab7 exhibited a significantly higher percentage of overlap (about 53%) compared to the other marker combinations (*Figure 3C*). An overlap of Rab5 and Rab7 has also been found in mammalian cells where it has been attributed to a temporary phenomenon during the maturation of EE to LE (*Podinovskaia et al., 2021*; *Rink et al., 2005*; *van der Beek et al., 2022*).

Interestingly, TbRab7 exhibited a higher degree of colocalization with TbRab11 (45%) than TbRab5A showed with TbRab11 (35%). This observation could indicate a functional polarization within the FP. TbRab11 is known to decorate disk-shaped structures called EXCs, which transport the recycled VSG coat back to the cell surface. *Boehm et al., 2017* report that in the FP endocytic and exocytic sites are in close proximity but do not overlap. We further suggest that the fusion of EXCs with the FP membrane and clathrin-mediated endocytosis take place on different sites of the pocket. This disparity explains the lower colocalization between TbRab11 and TbRab5A. It is worth noting that the mechanism by which Golgi-derived membrane and cargo are transported to the FP in trypanosomes remains unclear. In our view, the most plausible scenario involves feeding into the rapid endocytic recycling pathway, which is further supported by the absence of tubular carriers as observed in mammals.

Quantitative colocalization analysis by 3D IF provides statistically valid results, however, has the obvious limitation that it lacks ultrastructural information. Therefore, we analysed the localization of the different TbRab proteins on cryo-electron micrographs. This finally proved that different TbRab markers can be found on the same membrane structures and argues against the existence of distinct early, late, and recycling endosomal compartments in *T. brucei* (*Figures 4–6*). An alternative explanation for the two marker proteins residing on the same membrane would be the endosome maturation model, which has been proposed in mammals (*Poteryaev et al., 2010*; *Rink et al., 2005*; *Stoorvogel et al., 1991*). In short, EE undergo maturation and transform into late endosomes through a series of changes. This maturation process involves the acquisition of new membrane proteins and enzymes that among others facilitate a decrease in the pH value. Ultimately, the late endosomes fuse with lysosomes, leading to cargo degradation (reviewed in *Podinovskaia and Spang, 2018*).

EM cryosections provide high-resolution 2D information. To gain volume information, we resorted to an extremely demanding technique that combines ultrastructural 3D information with immunodetection. 3D Tokuyasu has not been applied to trypanosomes and the utilization of 3D immunogold has not been used on endosomes in any system. The Tokuyasu tomograms clearly showed the connection of circular cisternae and sheet-like structures in 3D (*Figure 8*). The decoration with different endosomal marker proteins demonstrated the existence of functional subdomains within a giant endosomal membrane system convincingly, and finally offered an explanation to the fundamental question of how trypanosomes can combine endosomal functionality with very fast membrane recycling.

Our work has certainly raised more questions than it has answered. One puzzling aspect is the presence of extended sheet-like endosomal structures that display sparse or no labelling of TbRab proteins. This is particularly intriguing when considering the substantial membrane area covered by these sheets. It is tempting to speculate that the absence of canonical markers correlates with a lack of endosomal function. We postulate that these sheets represent 'endosomal highways' that support fast flow of cargo by facilitated diffusion. This may allow for the efficient transport of VSG and fluid-phase cargo between distinct functional endosomal subdomains, which can be in close proximity or at a distance of microns apart. Consequently, the slow vesicular trafficking process in trypanosomes could be limited to CCVs, originating either from the FP or the rims of fenestrated sheets and cisternae, as well as to Rab11-positive EXCs.

Interestingly, we did not find any structural evidence of vesicular retrograde transport to the Golgi. Instead, the endosomal 'highways' extended throughout the posterior volume of the trypanosomes

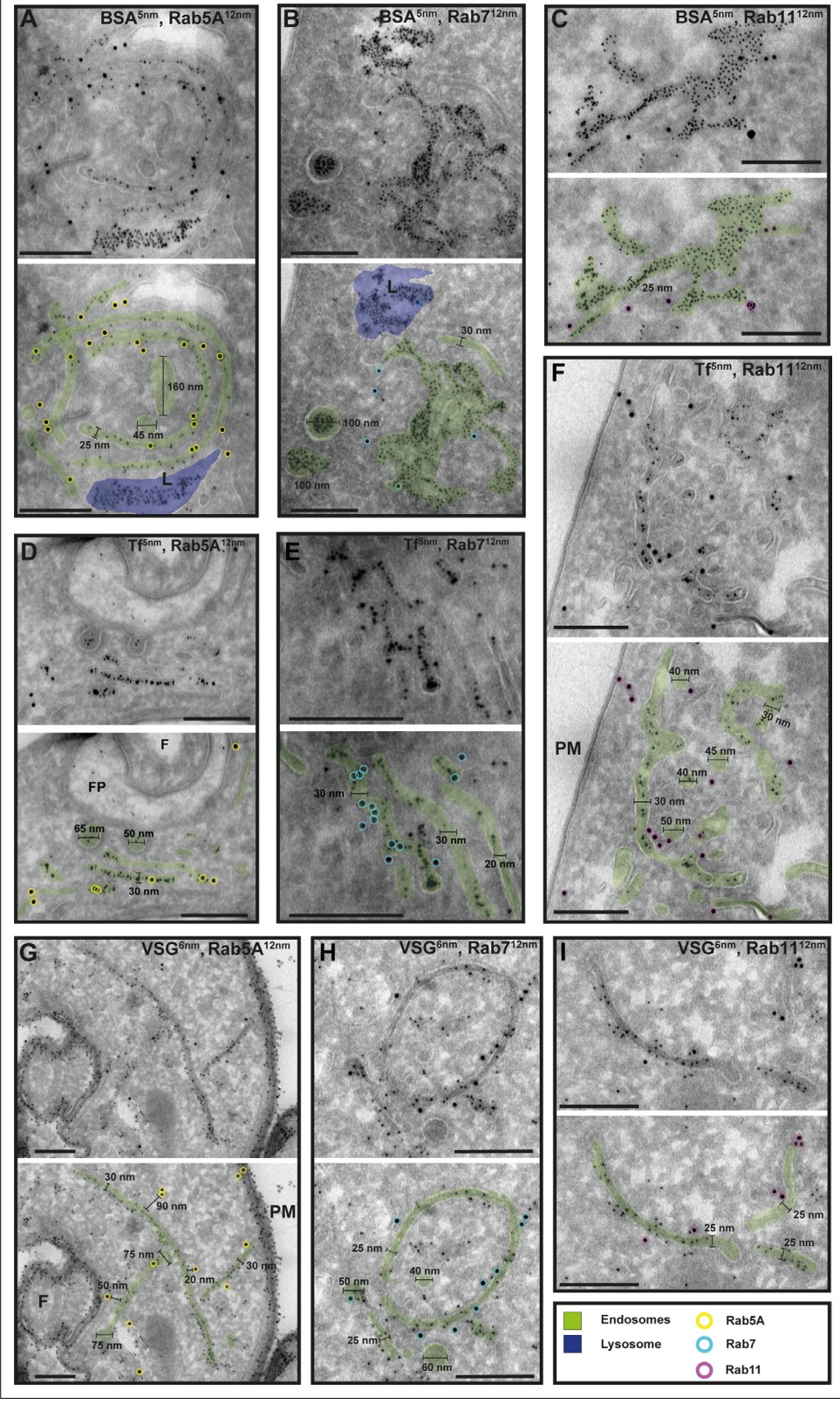

**Figure 7.** Cargo uptake and anti-VSG immunogold labelling confirm the endosomal identity. (**A–I**) Electron micrographs of cryosections labelled with anti-TbRab5A (yellow), anti-TbRab7 (cyan), or anti-TbRab11 (magenta) antibodies and 12 nm gold-conjugated secondary antibodies. For each panel, an annotated and an unedited version of the image is presented. Endosomes are highlighted in green, the lysosome in blue. Parasites were

*Figure 7 continued on next page*

*Figure 7 continued*

incubated with 5 nm gold-conjugated bovine serum albumin (BSA) (**A–C**) or transferrin (Tf) (**D–F**), prior to fixation, sectioning, and immunolabelling. BSA and transferrin cargo were observed in vesicles, endosomes, and lysosome. (**H, I**) Cells were labelled with anti-VSG antibodies and 6 nm gold-conjugated secondary antibodies. (F) Flagellum, (FP) flagellar pocket, (PM) plasma membrane, (L) lysosome. Scale bars: 200 nm. The displayed images are representative examples drawn from over 50 images generated in three distinct labelling experiments.

approaching the trans-Golgi interface. It is highly plausible that this region represents the convergence point where endocytic and biosynthetic membrane trafficking pathways merge. A comparable merging of endocytic and biosynthetic functions has been described for the trans-Golgi network (TGN) in plants. Different marker proteins for EE and RE were shown to be associated and/ or partially colocalized with the TGN suggesting its function in both secretory and endocytic pathways (reviewed in *Minamino and Ueda, 2019*). As we could not find structural evidence for the existence of a TGN we tentatively propose that trypanosomes may have shifted the central orchestrating function of the TGN as a sorting hub at the crossroads of biosynthetic and recycling pathways to the endosome. Although this is a speculative scenario, it is experimentally testable.

In summary, we confirmed the endosomal identity of the analysed membranes with the aid of cargo uptake assays (*Figures 1, 2, 7*), transgenic cell lines (*Figures 2 and 3*) as well as immuno labelling (*Figures 3–8*) combined with various light and electron microscopy methods. In the process, we ensured that all examined membranes belonged to the endosomal system and we did not, for example, mischaracterize ER membranes as endosomal membranes. All our results suggest that the endosomal apparatus consists of a continuous membrane system with TbRab5A-, TbRab7-, or TbRab11-positive subdomains that can be dynamically formed (*Figure 9*). Thus, the overall architecture of the trypanosome endosomes represents an adaptation that has facilitated the extreme speed of plasma membrane recycling observed in these organisms.

A partially continuous endosomal membrane structure has been proposed in the South American parasite *Trypanosoma cruzi* (*de Alcantara et al., 2018*; *Porto-Carreiro et al., 2000*), however, no endosome markers were used in that study to corroborate their findings. In the single-celled parasite *Giardia*, the peripheral vesicles that constitute the endosomal organelles form an 'interconnected network' (reviewed in *Benchimol and de Souza, 2022*), while the endosome-like compartment in *Toxoplasma* is likely involved in both endocytic and exocytic trafficking (reviewed in *McGovern et al., 2021*). In *Leishmania,* the endosomal system has not been fully characterized but seems to be under physical tension (reviewed in *Ansari et al., 2022*). Thus, parasitic protozoa might feature interesting variations of the common cell biological process of endocytosis. Furthermore, these variations might extend to many other organisms beyond the extensively studied, yet not entirely representative model organisms. The emergence of technologies such as CRISPR and single-cell analyses now enables the systematic exploration of the vast diversity of eukaryotic cells. Therefore, the systematic investigation of the phylogeny of cell biological inventions could be a logical next step.

## Materials and methods
### Cell culture and cell lines

*T. brucei* BSF parasites (Lister strain 427, antigenic type MITat 1.2, clone 221) were cultured in HMI-9 medium (*Hirumi and Hirumi, 1989*) supplemented with 10% heat-inactivated foetal calf serum (Sigma-Aldrich), 100 U/ml penicillin, and 0.1 mg/ml streptomycin. The cells were maintained at 37°C and 5% $CO_2$ split at regular intervals to keep the population densities below $1 \times 10^6$ cells/ml. Population density was monitored using a Z2 Coulter Counter (Beckman Coulter). The monomorphic '13–90' parasites expressing the T7 RNA polymerase and tetracycline repressor were cultivated in the presence of 5 µg/ml hygromycin and 2.5 µg/ml G418 (*Wirtz et al., 1999*). The transgenic BSF '2T1' cells which constitutively express a T7 RNA polymerase and tetracycline repressor were cultivated in the presence of 2.5 µg/ml phleomycin and 1 µg/ml puromycin (*Alsford et al., 2005*). The RNAi cell lines (RNAi-TbRab5A, RNAi-TbRab7, and RNAi-TbRab11) were based on the 2T1 cell line and cultivated in the presence of 5 µg/ml hygromycin and 2.5 µg/ml phleomycin. The transgenic EP1::GFP cells were based on MITat1.2 expressing wild-type cells and cultivated in the presence of 5 µg/ml hygromycin and 15 µg/ml G418 (*Engstler and Boshart, 2004*). The transgenic single marker (SM) cell line (*Wirtz*

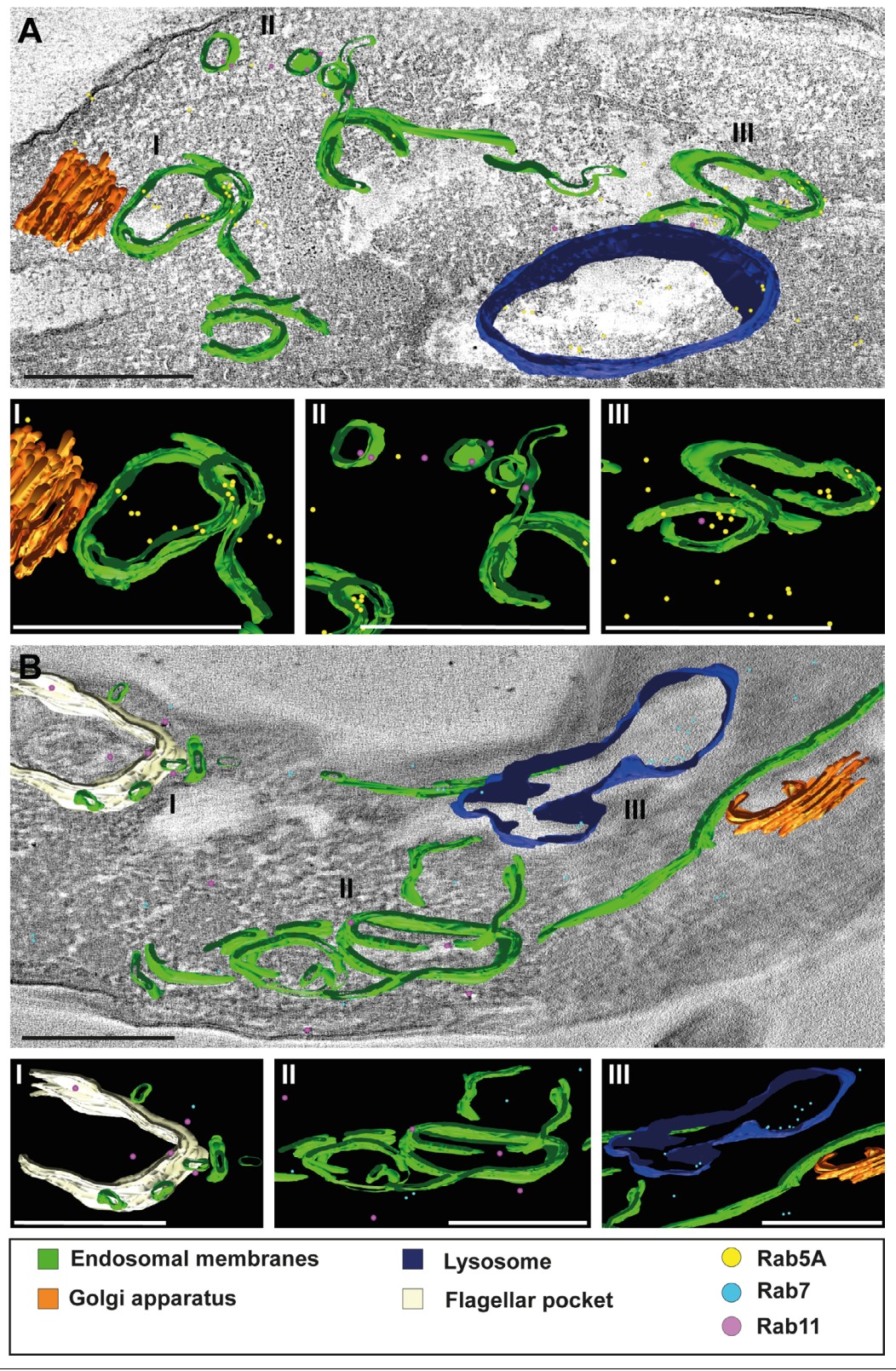

**Figure 8.** 3D Tokuyasu demonstrated continuous endosomal membranes in cryosections. (**A, B**) Tomographic reconstruction of the endosomal apparatus. (**I–III**) Isolated view of different organelle structures. Cryosections (250 nm) were labelled with anti-TbRab antibodies and 6 or 12 nm gold-coupled secondary antibodies. Endosomes are shown in green, lysosomes in blue, the flagellar pocket in white, the Golgi apparatus in orange. Gold

*Figure 8 continued on next page*

*Figure 8 continued*
particles are indicated as coloured spheres; TbRab5A (yellow), TbRab7 (cyan), or TbRab11 (magenta). Scale bars: 500 nm. Movies corresponding to panels A and B can be found here: (**A**) *Figure 8—video 1*, *Figure 8—video 2*; (**B**) *Figure 8—video 3*, *Figure 8—video 4*. The provided tomograms serve as a representative sample among the 35 tomograms that were recorded.

The online version of this article includes the following video(s) for figure 8:

**Figure 8—video 1.** z-stack of electron tomogram from *Figure 8A*.
https://elifesciences.org/articles/91194/figures#fig8video1

**Figure 8—video 2.** 3D model of electron tomogram from *Figure 8A*.
https://elifesciences.org/articles/91194/figures#fig8video2

**Figure 8—video 3.** z-stack of electron tomogram from *Figure 8B*.
https://elifesciences.org/articles/91194/figures#fig8video3

**Figure 8—video 4.** 3D model of electron tomogram from *Figure 8B*.
https://elifesciences.org/articles/91194/figures#fig8video4

---

*et al., 1999*) which constitutively express a T7 RNA polymerase and tetracycline repressor were cultivated in the presence of 2.5 µg/ml G418. The EP1::Halo-tagged cell line was based on SM cells and was cultivated in the presence of 2.5 µg/ml phleomycin and 2.5 µg/ml G418.

## Generation of RNAi cell lines

The RNAi target sequences were selected by entering the target gene ORF into the RNAit webapp (*Redmond et al., 2003*). The selected primer pairs flanked the base pairs 29–614 (TbRab5A), 134–630 (TbRab7), or 67–496 (TbRab11) and encoded the attB-sequence at the 5′ end for the ligation into the pGL2084 plasmid (*Jones et al., 2014*). The plasmids were linearized with AscI and 10 µg were transfected into $3 \times 10^7$ mid-log phase bloodstream 2T1 cells (*Alsford et al., 2005*) using Amaxa Basic Parasite Nucleofector Solution 1 using the X-001 program of an Amaxa Nucleofector II (Lonza, Switzerland). Clones were selected by growth in medium containing phleomycin (2.5 µg/ml) and hygromycin (5 µg/ml).

## Generation of EP1::Halo cell line

The insert for the EP1::HaloTag cell lines was generated in resemblance to the GFP::EP1 cell line from *Engstler and Boshart, 2004*. For this, the Halo tag was inserted in between the N-terminal EP1 globular domain and the C-terminal EP1 stalk domain. The three fragments (EP1 globular domain, Halotag, EP1 stalk domain) were amplified using PCR. For the EP1 fragments, the GFP::EP1 plasmid was used (*Engstler and Boshart, 2004*). While amplifying the globular domain, an internal BamHI restriction site was removed by silent point mutation. The Halo tag sequence was amplified from the pFC20A (HaloTag) T7 SP6 Flexi vector (Promega, USA). All amplified fragments were then combined using Gibson assembly (Gibson assembly cloning kit, NEB) with 20 base pair overhangs. This construct was cloned into pJet1.2blunt and later excised using HindIII and BamHI for ligation into the pLew100v5 expression vector (*Wirtz et al., 1999*). For transfection of the parental, SM cell line, 10 µg of the NotI linearized plasmid was transfected. Clones were selected by growth in medium containing phleomycin (2.5 µg/ml) and G418 (2.5 µg/ml). Primers used: forward primer globular domain (AGTAAAATTCACAAGCTTGG ATGGCACCTCGTTCCCTTTATCTGCTCGCTGTTCTTCTGTTCAGCGCGAACCTCTTCGCTGGCGTGGGATTTGCCGCAGCCGCTGAAGGACCAGAAGACA), reverse primer globular domain (CCGATTTCTGCCAT TCTAGAGGTGGCGACCGGCCGGTGAATCCCGCCGACCTTGGTTCCCTTCTCGCCTTTGCCTTTGCCTCCCTTAGTAAGACCCTTGTCTTCTGGTCCTTCAGCGGCT), forward primer Halo tag (TCTAGA ATGGCAGAAATCGGTACTGGCTTTCCATTCG), reverse primer Halo tag (TCAGTGCCATTGGTATCGTCCTTAATT AACCCGGAAATCTCCAGAGTAGACAGCC), forward primer stalk domain (TTAATTAA G GACGATACCAATGGCACTGACC), reverse primer stalk domain (AAAAGCCAACTAAATGGGCA GGATCC TTAGAATGCGGCAACGAGAGC).

## Growth curves

Cells were seeded with a starting concentration below $1 \times 10^6$ cells/ml in a volume of 10 ml and divided into two 2 ml aliquots in separate wells of a 24-well plate. Tetracycline was added to a final

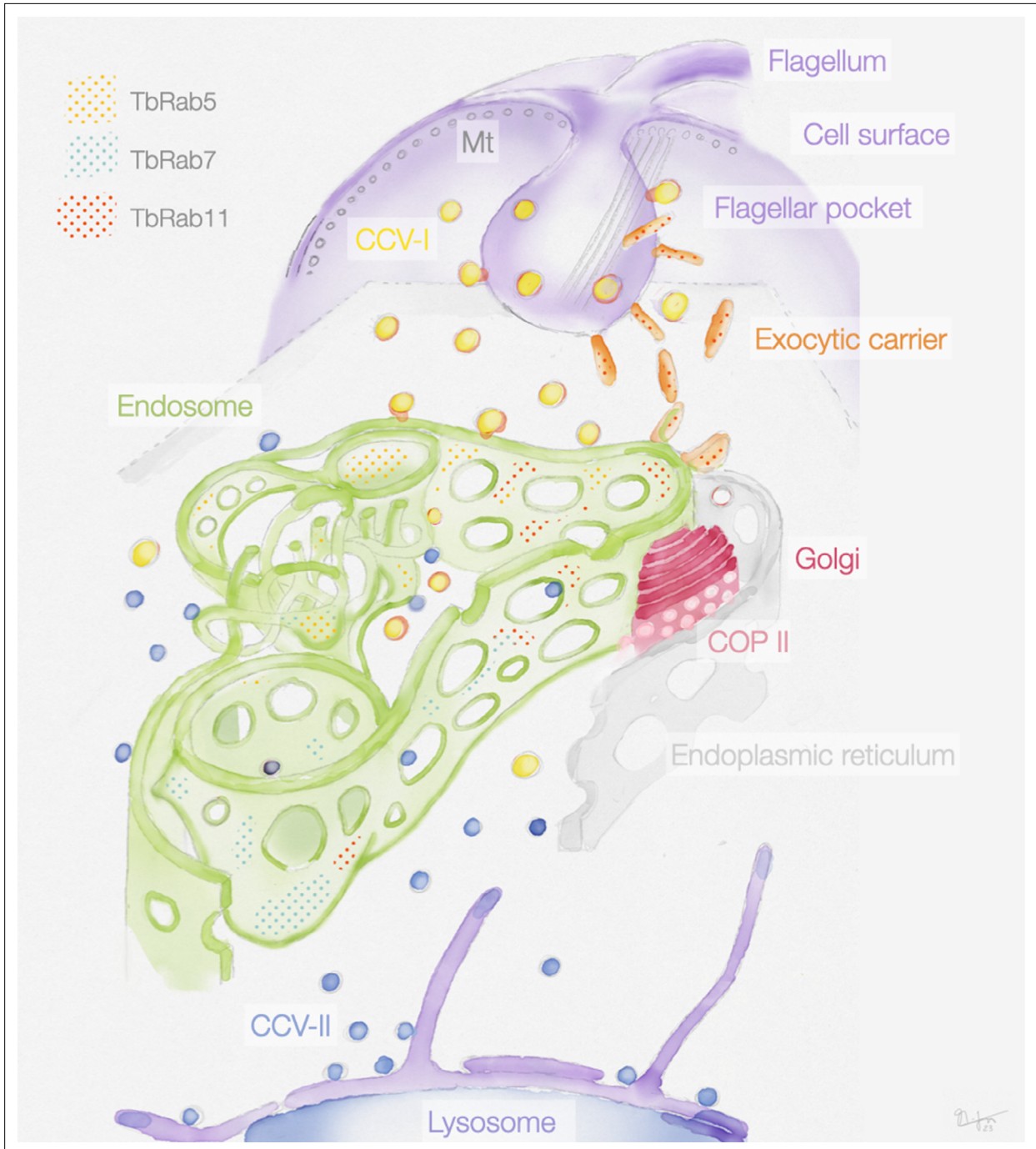

**Figure 9.** Schematic representation of the endosomal system in *T. brucei*. The endosome is marked by the presence of small GTPases of the Rab family: Rab5A (yellow dots), Rab7 (cyan dots), and Rab11 (red dots). Class I clathrin-coated vesicles (CCV I), class II clathrin-coated vesicles (CCV II), vesicles coated with coat protein (COP) II.

concentration of 1 µg/ml in one well to induce RNAi or EP1::Halo expression. The population density of the non-induced and induced cells was determined every 24 hr over a time course of 96 hr, using a Z2 Coulter Counter (Beckman Coulter). Depletion of the target protein was confirmed by immunoblotting of whole-cell lysates (Supplements 2).

## Cargo uptake

Per assay the same number of cells ($1 \times 10^7$ for light microscopy, $1 \times 10^8$ for electron microscopy) were harvested by centrifugation ($1500 \times g$, 10 min, 4°C). After that, cells were washed with 1 ml

trypanosoma dilution buffer (TDB; 5 mM KCl, 80 mM NaCl, 1 mM MgSO$_4$, 20 mM Na$_2$HPO$_4$, 2 mM NaH$_2$PO$_4$, 20 mM glucose, pH 7.6), harvested by centrifugation (1500 × $g$, 2 min, 4°C) and the supernatant was discarded. Next, the cells were resuspended in 100 µl TDB and incubated with either dextran-Alexa488 (light microscopy), transferrin-gold conjugates, or BSA-gold conjugates (electron microscopy) for 15 min at 37°C or HRP/ferritin for 5 min at 37°C (electron microscopy). The final concentration of both dextran (Thermo Fisher cat. number: D22910) and HRP (horseradish peroxidase type II, Sigma) was 10 mg/ml. The final concentration of ferritin (horse spleen ferritin, type I, Sigma) was 50 mg/ml. Transferrin-gold (Cytodiagnostics) or BSA-gold (Alexa555-BSA-Au, UMC Utrecht) were added to a final OD = 5. The uptake reactions were stopped by adding 1 ml of the corresponding fixatives (4% formaldehyde and 0.2% glutaraldehyde for the dextran uptake, 2% formaldehyde and 0.2% glutaraldehyde for BSA and transferrin uptake, 2.5% glutaraldehyde for HRP uptake, 2% formaldehyde and 1% glutaraldehyde for ferritin uptake). Cells were fixed for 20 min on ice followed by 30 min at room temperature (RT)(for dextran uptake), 1 hr at RT (for HRP and ferritin) or 2 hr at RT (for BSA and transferrin uptake). Following this, cells were treated either as described in the immunofluorescence section or as explained in the sample embedding and immuno-electron microscopy section.

## EP1::Halo cell labelling

Cells were harvested (1400 × $g$, 10 min, 37°C) and washed in 1 ml TDB. The cell pellet was resuspended in 100 µl fresh TDB and HaloTag ligand either in-house labelled Alexa647 ligand or Halo-Tetramethylrhodamine (TMR, Promega, USA) was added to a final concentration of 500 nM (Alexa647) or 5 nM (TMR). The labelling reaction was performed for 15 min at 37°C and stopped by the addition of 4% formaldehyde in phosphate-buffered saline (PBS).

## Immunofluorescence

Cells were harvested (1400 × $g$, 10 min, 37°C), washed in ice-cold TDB, and fixed with 4% formaldehyde overnight (4°C) or for 1 hr at RT. Fixed samples were washed in 1 ml PBS, harvested (1400 × $g$, 90 s, RT) and 1 × 10$^6$ cells were added to each Ø 12 mm coverslip coated with 0.01% poly-L-lysine (>20 min, RT). The cells were attached to the coverslips by centrifugation (750 × $g$, 1 min, RT) and attachment was confirmed visually. The attached cells were then permeabilized in 0.25% Triton X-100 in PBS (5 min, RT) and blocked in 3% (wt/vol) BSA in PBS (30 min, RT). Sequentially, coverslips were incubated with primary antibodies (1 hr, RT), washed with PBS (3 × 5 min), and incubated once more with secondary antibodies (1 hr, RT, in the dark). Finally, samples were washed in PBS (3 × 5 min), rinsed in ddH$_2$O, carefully dried, and mounted on glass slides using ProLong Diamond Antifade (Life Technologies). Antibodies were diluted in 1% (wt/vol) BSA in PBS as follows: rat anti-TbRab5A 1:250, guinea pig anti-TbRab7 1:250, rabbit anti-TbRab11 1:250, goat anti-rat conjugated with Alexa Fluor 647 1:500, goat anti-guinea pig conjugated with CF568 1:500, goat anti-rabbit conjugated with Alexa Fluor 405 1:500.

## Expansion microscopy

The proExM method was performed as described in *Tillberg et al., 2016* with some modifications. Approximately 1 × 10$^7$ cells were harvested by centrifugation (1500 × $g$, 10 min, 4°C), then washed with 1 ml TDB, harvested by centrifugation (1500 × $g$, 2 min, 4°C) again, and the supernatant was discarded. Next, the cells were resuspended in 100 µl TDB and incubated with dextran-Alexa488 for 15 min at 37°C. The final concentration of dextran (Thermo Fisher cat. number: D22910) was 10 mg/ml. The reaction was stopped by adding 1 ml 4% formaldehyde and 0.2% glutaraldehyde in PBS. Cells were fixed for 20 min on ice followed by 30 min at RT. After the fixation, cells were adhered to 12 mm coverslips. For anchoring of proteins, a solution of 5 mg acryloyl-X (AcX (A2 0700, Invitrogen), dissolved in freshly opened 500 µl anhydrous dimethyl sulfoxide (DMSO) (D12345, Invitrogen), was prepared. A 100-µl drop of AcX 1:100 in PBS was placed on parafilm in a humidity chamber, the coverslips were inverted and incubated at room temperature for 16 hr. Coverslips were returned to a 24-well plate, washed twice for 5 min with PBS and stored in the fridge until the end of the day. For gelation, gelling solution was made mixing 100 µl monomer solution with 0.5 µl 10% (vol/vol) ammonium persulfate (APS) and 0.5 µl tetramethylethylenediamine (TEMED), vortexed and used straight away. A volume of 50 µl was placed on a piece of parafilm in a humidity chamber and the coverslips were inverted and incubated at 37°C for 1 hr. Monomer solution was prepared with sodium

acrylate 8.6% (wt/vol) (408220, Sigma), acrylamid 2.5% (wt/vol), *N-N'*-methylenebisacrylamide 1.5% (wt/vol) (M7279, Sigma), and NaCl 11.7% (wt/vol) (AM9760G, Ambion) in PBS. Sodium acrylate stock solution 33% (wt/vol) was prepared by dissolving 1.9 g in 5 ml of ddH$_2$O, 0.2 mm filtered and used freshly. After polymerization, coverslips with the gel stuck to it were transferred to a 6-well plate with 1 ml of digestion solution 0.5% Triton X-100, 1 mM ethylenediaminetetraacetic acid (EDTA)), pH 8 (AM9260G, Ambion), 50 mM Tris, pH 8 (AM9855G, Ambion) and 1 M NaCl (AM9760G, Ambion) freshly supplemented with 10 U/ml proteinase K (P8107S, New England Biolabs). Plates were slightly tilted to ensure the coverslip was completely covered and incubated at room temperature overnight. The next day, gels were detached from the coverslips and transferred to a Petri dish. After three rounds of expansion with 20 ml ddH$_2$O for 30 min each, gels were mounted in a poly-D-lysine coated imaging chamber (Nunc 155360 Lab-Tek II Chambered Coverglass with Cover, No. 1.5 Borosilicate Sterile) and imaged using a DMi8 widefield microscope.

## Fluorescence microscopy

Images were acquired using a DMi8 widefield microscope (Thunder Imager, Leica Microsystems, Germany) with a HCX PL APO CS objective (×100 oil immersion, NA 1.4, working distance 90 µm) and Type F Immersion Oil (refractive index = 1.518, Leica Microsystems, Germany). The microscope was operated using LAS-X software (Leica). Samples were illuminated with an LED8 light source (Leica) and 50% illumination power. Excitation light was selected by using following filter sets: 391/32 nm; 479/33 nm; 554/24 nm; 638/31 nm (Leica Microsystems, Germany). Emitted light was collected at 435/30 nm (DAPI and Alexa Fluor 405), 519/25 nm (Dextran-Alexa Fluor 488 and GFP), 594/32 nm (CF568 and TMR), and 669 nm (Alexa Fluor 647), respectively. The individual exposure times were as follows: DAPI, Alexa Fluor 405, CF568, Alexa Fluor 647 = 200 ms; GFP = 500 ms; Dextran-Alexa Fluor 488=2 s; TMR = 2 s. Differential interference contrast (DIC) was used to visualize cell morphology. 3D recording of each field of view was obtained using 48 z-slices (step size = 0.21 µm). Fields of view were selected based on DIC to subjectively select for areas containing enough cells and blind the user to the fluorescence signal of the protein markers of interest. Images were captured using a Leica K5 sCMOS camera (pixel size 6.5 µm). The LAS-X/Thunder integrated instant computational clearing algorithm was used to remove background signal, using a minimal structure size of 338 nm.

## Colocalization analysis

Widefield microscopy images were deconvolved using Huygens Essential version 21.10 (Scientific Volume Imaging B.V., Hilversum, Netherlands) and analyzed with IMARIS (9.9.1, Oxford Instruments). In brief, surfaces were rendered for the EP1 signal, and checked manually for accurate posterior localization in morphologically intact cells. These EP1 surfaces were used to define the ROI for colocalization analysis. The automatic thresholding function (*Costes et al., 2004*) was used to ensure objective thresholding for the colocalization analysis of the different TbRab marker protein combinations.

## *d*STORM imaging

*d*STORM imaging was performed on an inverted widefield microscope (Olympus, IX-71). HaloTag-Ligand Alexa Fluor 647 was excited by a 641-nm diode laser (Topica, iBeam smart 640S), a 405-nm diode laser (Coherent, Cube 405-100C) was used for DAPI excitation. Both Lasers were spectrally cleaned with cleanup filters (Chroma, laser clean up filter 640/10 and 405/20) and focussed on the back focal plane of a ×60 oil-immersion objective (Olympus, ×60 NA 1.45). Emission light was separated from excitation light using a dichroic beam splitter (Semrock, FF410/504/583/669) and further spectrally filtered by a long-pass filter (Semrock, 647 nm RazorEdge). Images were recorded on an EM-CCD camera chip (Andor, iXon DU-897) with a measured pixel size of 134 nm. *d*STORM imaging was performed in a buffer of 100 mM MEA (Sigma-Aldrich, #M6500) in PBS at pH 7.4. Time series of 15,000 frames at an exposure time of 20 ms were captured while irradiating the sample by highly inclined laminated optical sheet (HILO) illumination with an intensity of approx. 2 kW cm$^{-2}$. Super-resolved images were reconstructed with rapidSTORM 3.3 (*Wolter et al., 2012*) and further processed for illustrations in Fiji. Widefield images of transmitted light and DAPI staining were captured after *d*STORM. For illustrations, the scale of the widefield images was matched to the super-resolved *d*STORM images.

## SIM imaging

Structured illumination imaging was performed using a Zeiss Elyra 7 lattice SIM equipped with a ×63 C-Apochromat ×63/1.20 W (Zeiss) water immersion objective, four excitation lasers (405 nm diode, 488 nm OPSL, 561 nm OPSL, and 642 nm diode laser) and two sCMOS (PCO, pco.edge 4.2) cameras. Fourfold expanded gels were transferred onto PDL coated single well chamber slides (CellVis, #C1-1.5H-N). For excitation of DAPI the 405 nm laser diode was set to 10% output power, for excitation of Dextran-Alexa488 the 488 nm laser to 20%. A respective exposure time of 100 and 200 ms was set for the two channels. Z-stacks were captured in 3D Leap mode with optimal step intervals and appropriate filter sets (BP 420–480 + LP 655 and BP 495–550 + BP 570–620 for the DAPI- and 488-channel, respectively). SIM imaging was performed by capturing 13 phase shifts of the lattice SIM pattern per slice. Super-resolved images are processed in ZEN 3.0 SR FP2 (black) (Zeiss, Version 16.0.10.306) with SIM and SIM² processing modules.

## Sample embedding and preparation for electron microscopy

For Tokuyasu immuno-EM, sample preparation and sectioning were performed as previously described (*Engstler et al., 2004*; *Grünfelder et al., 2003*; *Möbius and Posthuma, 2019*; *Slot and Geuze, 2007*). In short, $1 \times 10^8$ cells were harvested ($1500 \times g$, 10 min, 37°C) and washed with 1 ml TDB. Cells were fixed by addition of 4% formaldehyde and 0.4% glutaraldehyde in 0.2 M PHEM buffer (120 mM piperazine-N,N'-bis(2-ethanesulfonic acid (**P**IPES), 50 mM 4-(2-hydroxyethyl)-1-piperazineethanesulfo nic acid) (**H**EPES), 20 mM ethylene glycol-bis(β-aminoethyl ether)-N,N,N',N'-tetraacetic acid (**E**GTA), 4 mM **M**gCl$_2$ adjusted to pH 6.9 with 1 M NaOH) 1:1 to TDB. After 5 min the fixatives were replaced with 2% formaldehyde and 0.2% glutaraldehyde in 0.1 M PHEM for 2 hr at RT. Fixatives were removed by washing with 1 ml TDB and quenched with PBS + 0.15% (wt/vol) glycine. Cells were pelleted and resuspended in 200 µl 12% (wt/vol) gelatine (Rousselot 250 LP30) in 0.1 M PHEM at 37°C and pelleted again. The pellets were solidified on ice, cut into smaller blocks, and infused with 2.3 M sucrose in 0.1 M PHEM overnight at 4°C. The blocks were mounted on metal pins and stored in liquid nitrogen. The gelatine-embedded cells were cryosectioned to 55 or 250 nm thick sections at −120 or −100 °C, respectively, with a DiATOME diamond knife in a Leica ultracut cryomicrotome (UC7). Sections were collected and deposited on formvar- and carbon-coated grids using 2.3 M sucrose and 1.8% (vol/vol) methylcellulose (MC; Sigma M6385 25 centipoises) mixed 1:1.

For Epon embedding, sample preparation and sectioning were performed as previously described (*Langreth and Balber, 1975*; *Webster, 1989*). In short, chemically fixed cells were pelleted ($1500 \times g$, 10 min, RT) and washed 6× in 0.12 M sucrose, 0.1 M cacodylate, pH 7.4. They were postfixed in 1% (wt/vol) osmium tetroxide, stained *en bloc* in 1% (wt/vol) uranyl acetate (UA) in 50 mM sodium maleate buffer (pH 5.1), dehydrated with ethanol and embedded at 60°C in epoxy resin. Cells incubated with HRP were treated, in suspension, with DAB in 50 mM Tris–HCl (pH 7.6) containing 0.01% (vol/vol) osmium tetroxide for 5 min prior to postfixation with 1% osmium tetroxide. The epon-embedded cells were sectioned to 250 nm thick sections at RT.

## Immuno-electron microscopy

Grids with sections facing down were incubated in a 24-well plate with 2 ml PBS at 37°C for 30–60 min to remove gelatine, 2.3 M sucrose and 1.8% MC mixture. Afterwards, the grids were washed on ~100 µl drops of PBS + 0.15% (wt/vol) glycine and PBS on parafilm at RT (both 3 × 5 min). The standard volume for all non-antibody-related steps was 100 µl. Next, blocking was performed in PBS + 0.1% (vol/vol) acetylated BSA (BSA-c, Aurion) + 0.5% (wt/vol) cold fish-skin gelatine (FSG, Sigma) for 30 min. The following primary antibody incubation was done for 1–2 hr in PBS + 0.1% BSA-c + 0.5% FSG. The used antibody dilutions were rat anti-TbRab5A 1:50, guinea pig anti-TbRab7 1:50, rabbit anti-TbRab11 1:25, and rabbit anti-VSG 1:50. After that, grids were washed 5 × 5 min with PBS + 0.1% BSA-c + 0.5% FSG and incubated with corresponding gold-coupled secondary antibodies for 1–2 hr (diluted 1:10 in PBS + 0.1% BSA-c + 0.5% FSG). Further washing steps with PBS +0.1% BSA-c + 0.5% FSG (3 × 5 min) and PBS (3 × 5 min) followed before all remaining antibodies were fixed with 1.25% glutaraldehyde for 5 min at RT. After three washing steps with ddH$_2$O to remove phosphate from the grids, sample contrast was increased by 5 min incubation on 2% (wt/vol) UA in water (pH 7) and 7-min incubation on 1.8% (vol/vol) MC/0.3% (wt/vol) UA in water (pH 4) (both on ice). Grids were

retrieved with a metal ring and excess MC/UA was removed with filter paper. Grids were dried for at least 30 min at RT in the metal ring.

## Electron microscopes

JEOL JEM-2100 TEM operated at 200 kV with camera system: TVIPS F416 4k × 4k.

JEOL JEM-1400 Flash scanning transmission electron microscope operated at 120 kV with camera system: Matataki Flash 2k × 2k.

FEI Tecnai TF30 TEM operated at 300 kV with camera system: Gatan US4000 4k × 4k CCD.

The software Serial EM was used for tilt series acquisitions of electron tomograms. The subsequent reconstruction and segmentation were done using the IMOD software package (*Kremer et al., 1996*). For 2D electron micrographs pseudo colouring and highlighting was done using Adobe Illustrator 2022.

## Immunoblotting

Trypanosomes were resuspended and lysed in protein sample buffer (60 mM Tris–HCl, pH 6.8, 2% (wt/vol) sodium dodecyl sulfate (SDS), 10% (vol/vol) glycerol, 0.002% (wt/vol) bromophenol blue, 1% (vol/vol) β-mercaptoethanol) to yield equivalents of $2 \times 10^5$ cells/µl. For quantification of TbRab proteins by western blot 20 µl of the protein sample was resolved on 12.5% SDS–polyacrylamide gel electrophoresis (PAGE) gels and transferred onto nitrocellulose membrane. The membranes were blocked for 1 hr at room temperature with 5% (wt/vol) milk powder in PBS. Next, the membranes were incubated in polyclonal primary antibodies diluted in PBS containing 1% (wt/vol) powdered milk and 0.1% (vol/vol) Tween 20, anti-TbRab 1:5000 and monoclonal mouse anti-TbPFR (L13D6) 1:20 (*Kohl et al., 1999*). After wash steps, the membranes were incubated in IRDye 800 CW (Li-Cor) labelled either goat anti-rat, donkey anti-guinea pig, or donkey anti-rabbit and IRDye 680 LT (Li-Cor) labelled goat anti-mouse secondary antibodies diluted 1:10,000 in PBS containing 1% (wt/vol) milk powder and 0.1% (vol/vol) Tween 20. Images were acquired and quantified with the Li-Cor Odyssey system.

## Recombinant protein expression and purification for antibody generation

The TbRab5A (Tb927.10.12960), TbRab7 (Tb927.9.11000), TbRab11 (Tb927.8.4330) open reading frames were amplified from *Trypanosoma brucei brucei* strain Lister 427 genomic DNA by PCR. The PCR product was digested with NdeI and XhoI restriction enzymes and ligated into the pET21-a expression vector, which encodes an N-terminal His6 tag. The plasmid was used to perform a chemical transformation into *E. coli* strain Rosetta II (DE3) pLysS. Individual colonies were subsequently grown at 37°C in presence of 100 µg/ml ampicillin to an OD600 of ~0.8. Recombinant protein expression was induced by the addition of 50 µM isopropyl-β-D-1-thiogalactopyranoside (IPTG) and the cells were incubated for 4 hr at 37°C with shaking. Cells were harvested by centrifugation (3000 × *g*, 10 min, 4°C). The pellet was resuspended in 10 ml lysis buffer (100 mM $NaH_2PO_4$, 10 mM Tris/HCl, 8 M Urea, pH 8.0) and incubated for 1 hr at RT on. Final lysis was then achieved with sonication on ice, using 6 cycles for 10 s with MiniTip step 7 (Sonifier, B-12, Branson Ultrasonics). Lysates were clarified by centrifugation (10,000 × *g*, 30 min, RT) and incubated for 1 hr with Ni-NTA His bind resin (Merck) rolling at RT. Afterwards, the resin was transferred to a HiTrap column (GE Healthcare) and washed with 30 ml lysis buffer and 30 ml buffer B (100 mM $NaH_2PO_4$, 10 mM Tris/HCl, 8 M urea, pH 6.3). Next, the bound proteins were eluted with 1 × 2 ml buffer C (100 mM $NaH_2PO_4$, 10 mM Tris/HCl, 8 M urea pH, 5.9), 1 × 2 ml buffer D (100 mM $NaH_2PO_4$, 10 mM Tris/HCl, 8 M urea pH 4.5), and 1 × 2 ml buffer E (100 mM $NaH_2PO_4$, 10 mM Tris/HCl, 8 M urea, pH 4.5 + 250 mM imidazole). All elutions were then separately dialysed overnight against a 20-mM phosphate buffer (2.7 mM $Na_2HPO_4$, 17.3 mM $NaH_2PO_4$) at 4°C. The purification success was validated by 15% SDS–PAGE and proteins were sent to different companies for antibody generation.

## Antibody generation

Recombinant full-size proteins were used to immunize different animals (Pineda, Berlin, Germany; Davids Biotechnology, Regensburg, Germany; Eurogentec, Seraing, Belgium). The resulting antisera were affinity purified against the immobilized recombinant protein. The resulting antisera from two

rats (anti-TbRab5A) showed no need for affinity purification. All antibodies were validated for selectivity through immunoblots of the corresponding RNAi cell line (*Figure 3—figure supplement 1*).

## HaloTag ligand labelling

HaloTag-Ligand-Amine (Promega, #P6741) was reconstituted in N,N-dimethylformamid (DMF; Sigma-Aldrich, #70547) to a concentration of 10 g/l. 250 µg HaloTag-Ligand Amine was conjugated to Alexa Fluor 647-NHS (Thermo Fisher, #A20006) by adding 150 µg of NHS-dye from a 10 g/l stock solution in DMF. The total reaction volume was adjusted to 120 µl with DMF, then DIPEA Base (Sigma-Aldrich, #387649) was added at 1:100. The reaction was incubated under gentle agitation and protected from light at room temperature for 3 hr. The reaction mixture was purified by reverse-phase high-performance liquid chromatography (Jasco) using a biphenyl column (Kinetex, #00F-4622-E0) with a solvent gradient of 0–60% (vol/vol) acetonitrile (Sigma-Aldrich, #34851) in 0.1% trifluoroacetic acid (TFA; Sigma-Aldrich, #T6508) in water run over 30 min. Collected product fractions were dried in a speed-vac (Thermo Fisher, #SPD111V) and dissolved in DMSO.

## Acknowledgements

We thank the Cell Microscopy Core of UMC Utrecht for their workshop 'Cryosectioning and immuno-electron microscopy' and the opportunity to learn the Tokuyasu technique. Manfred Alsheimer provided protocols and advice with recombinant protein expression and antibody generation. Daniela Bunsen and Claudia Gehring-Höhn provided help with the EM benchwork and assistance with the electron microscope. Bernardo Gabiatti, Janna Eilts, Marcel Rühling, and Tobias Kunz provided protocols and advice on expansion microscopy. We are grateful to Brooke Morriswood for essential discussions. We express our gratitude to the eLife reviewers for dedicating their time and effort to evaluate the manuscript. We sincerely value all the insightful comments and suggestions, particularly those related to obtaining tomograms of unprocessed cells for use as a control. These recommendations played a crucial role in enhancing the overall quality of the manuscript. Funding: ME was supported by DFG grants EN305, SPP1726 (Microswimmers—From Single Particle Motion to Collective Behaviour), GIF grant I-473-416.13/2018 (Effect of extracellular *Trypanosoma brucei* vesicles on collective and social parasite motility and development in the tsetse fly) and GRK2157 (3D Tissue Models to Study Microbial Infections by Obligate Human Pathogens), the EU ITN Physics of Motility, and the BMBF NUM Organo-Strat. ME is a member of the Wilhelm Conrad Röntgen Center for Complex Material Systems (RCCM). The transmission electron microscopes in Würzburg were funded by the Deutsche Forschungsgemeinschaft (DFG, German Research Foundation) – 426173797 (INST 93/1003-1 FUGG) and 218894163. AB was supported by the Brazilian agency CAPES (program: CAPES/DAAD—Call No. 22/2018; process 88881.199683/2018-01). MJ was supported by GRK2157 and the European Research Council (ERC) under the European Union's Horizon 2020 research and innovation programme (grant agreement No. 835102) awarded to MS.

## Additional information

### Funding

| Funder | Grant reference number | Author |
| --- | --- | --- |
| Deutsche Forschungsgemeinschaft | EN305 | Markus Engstler |
| German-Israeli Foundation for Scientific Research and Development | I-473-416.13/2018 | Markus Engstler |
| Deutsche Forschungsgemeinschaft | 426173797 (INST 93/1003-1 FUGG) 218894163 | Christian Stigloher |
| Coordenação de Aperfeiçoamento de Pessoal de Nível Superior | 88881.199683/2018-01 | Alyssa Borges |

| Funder | Grant reference number | Author |
|---|---|---|
| HORIZON EUROPE European Research Council | 835102 | Markus Sauer |
| Deutsche Forschungsgemeinschaft | SPP1726 | Markus Engstler |

The funders had no role in study design, data collection, and interpretation, or the decision to submit the work for publication.

## Author contributions

Fabian Link, Conceptualization, Formal analysis, Validation, Investigation, Visualization, Methodology, Writing – original draft, Writing - review and editing; Alyssa Borges, Conceptualization, Investigation, Methodology, Writing – original draft; Oliver Karo, Stefan Sachs, Mary Morphew, Investigation; Marvin Jungblut, Elisabeth Meyer-Natus, Investigation, Methodology; Thomas Müller, Conceptualization, Investigation; Timothy Krüger, Formal analysis, Validation, Investigation; Nicola G Jones, Conceptualization; Markus Sauer, Funding acquisition, Project administration; Christian Stigloher, Conceptualization, Resources, Supervision; J Richard McIntosh, Conceptualization, Resources, Investigation; Markus Engstler, Conceptualization, Resources, Data curation, Formal analysis, Supervision, Funding acquisition, Validation, Investigation, Methodology, Writing – original draft, Project administration, Writing - review and editing

## Author ORCIDs

Fabian Link ⓘ http://orcid.org/0000-0002-9828-2012
Thomas Müller ⓘ http://orcid.org/0009-0005-0531-8338
Timothy Krüger ⓘ http://orcid.org/0000-0001-5544-1098
Markus Sauer ⓘ https://orcid.org/0000-0002-1692-3219
Christian Stigloher ⓘ http://orcid.org/0000-0001-6941-2669
Markus Engstler ⓘ http://orcid.org/0000-0003-1436-5759

Reviewer #2 (Public Review): https://doi.org/10.7554/eLife.91194.3.sa1
Reviewer #3 (Public Review): https://doi.org/10.7554/eLife.91194.3.sa2
Author response https://doi.org/10.7554/eLife.91194.3.sa3

---

# Additional files

## Supplementary files

• MDAR checklist

## Data availability

All raw data are uploaded to BioImage Archive and accessible via https://www.ebi.ac.uk/biostudies/bioimages/studies/S-BIAD1080.

The following dataset was generated:

| Author(s) | Year | Dataset title | Dataset URL | Database and Identifier |
|---|---|---|---|---|
| Link F, Engstler M | 2024 | Continuous endosomes form functional subdomains and orchestrate rapid membrane trafficking in trypanosomes | https://www.ebi.ac.uk/biostudies/bioimages/studies/S-BIAD1080 | BioImage Archive, 10.6019/S-BIAD1080 |

---

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
