## [Editor Report · eLife assessment]

This **important** study combines a range of advanced ultrastructural imaging approaches to define the unusual endosomal system of African trypanosomes. **Compelling** images reveal that, unlike a conventional set of compartments, the endosome in these protists forms a continuous membrane system with functionally distinct subdomains, as defined by canonical markers for early, late, and recycling endosomes. The findings compellingly support that the endocytic system in bloodstream stages has adapted to support remarkably high rates of membrane turnover necessary for immune complex removal and survival in the blood. This research is particularly relevant to those investigating infectious diseases

---

## [Referee Report · Reviewer #2 (Public Review)]

The authors suggest that the African trypanosome endomembrane system has unusual organisation, in that the entire system is a single reticulated structure. It is not clear if this is thought to extend to the lysosome or MVB. There is also a suggestion that this unusual morphology serves as a trans-(post)Golgi network rather than the more canonical arrangement.

The updated manuscript is significantly improved. I remain at slight odds with the author's push for the lack of generality as important, and the new cell biology that we have been on the verge of for decades. However, that is a scholarly issue and is not grounds for any further revision of the present manuscript.

---

## [Referee Report · Reviewer #3 (Public Review)]

Summary:

A key element in the ability of trypanosomes to evade the mammalian host's immune system is its high rate of endocytosis. This rapid turnover of its surface enables the trypanosome to 'clean' its surface removing antibodies and other immune effectors that are subsequently degraded. The high rate of endocytosis is likely reflected in the organisation of the endosomal system in these parasites. Here, Link et al., sought to address this question using a range of light and three-dimensional electron microscopy approaches to define the endosomal organisation in this parasite.

Before this study, the vast majority of our information about the make-up of the trypanosome endosomal system was from thin section electron microscopy and immunofluorescence studies, which did not provide the necessary resolution and 3D information to address this issue. Therefore, it was not known how the different structures observed by EM were related. Link et al., have taken advantage of the advances in technology and used an impressive combination of approaches at the LM and EM level to study the endosomal system in these parasites. This innovative combination has now shown the interconnected-ness of this network and demonstrated that there are no 'classical' compartments within the endosomal system, with instead different regions of the network enriched in different protein markers (Rab5a, Rab7, Rab11). Overall, the authors have achieved their aims, with results supporting their conclusions.

This is a well written manuscript in which the authors use an impressive range of approaches to address the organisation of the endosomal system. The authors have clearly demonstrated that trypanosomes have a large interconnected endosomal network, without defined compartments and instead shows enrichment for specific Rabs within this network. I appreciate their inclusion of how they used a range of different light microscopy approaches even though for instance the dSTORM approach did not turn out to be as effective as hoped.

The methodological impact of this work has the potential to be large, as the authors have introduced a range of advanced EM techniques for the study of trypanosomes. Moreover, the study of fundamental biological processes such as endosomal trafficking in divergent eukaryotes is important to define the limits within which this process operates.

---

## [Author Response]

The following is the authors’ response to the original reviews.

**eLife assessment**
This important study combines a range of advanced ultrastructural imaging approaches to define the unusual endosomal system of African trypanosomes. Compelling images show that instead of a distinct set of compartments, the endosome of these protists comprises a continuous system of membranes with functionally distinct subdomains as defined by canonical markers of early, late and recycling endosomes. The findings suggest that the endocytic system of bloodstream stages has evolved to facilitate the extraordinarily high rates of membrane turnover needed to remove immune complexes and survive in the blood, which is of interest to anyone studying infectious diseases.
**Public Reviews:**

**Reviewer #1 (Public Review):**
Summary:Bloodstream stages of the parasitic protist, *Trypanosoma brucei*, exhibit very high rates of constitutive endocytosis, which is needed to recycle the surface coat of Variant Surface Glycoproteins (VSGs) and remove surface immune complexes. While many studies have shown that the endo-lysosomal systems of T. brucei BF stages contain canonical domains, as defined by classical Rab markers, it has remained unclear whether these protists have evolved additional adaptations/mechanisms for sustaining these very high rates of membrane transport and protein sorting. The authors have addressed this question by reconstructing the 3D ultrastructure and functional domains of the *T. brucei* BF endosome membrane system using advanced electron tomography and super-resolution microscopy approaches. Their studies reveal that, unusually, the BF endosome network comprises a continuous system of cisternae and tubules that contain overlapping functional subdomains. It is proposed that a continuous membrane system allows higher rates of protein cargo segregation, sorting and recycling than can otherwise occur when transport between compartments is mediated by membrane vesicles or other fusion events.Strengths:The study is a technical tour-de-force using a combination of electron tomography, super-resolution/expansion microscopy, immune-EM of cryo-sections to define the 3D structures and connectivity of different endocytic compartments. The images are very clear and generally support the central conclusion that functionally distinct endocytic domains occur within a dynamic and continuous endosome network in BF stages.Weaknesses:The authors suggest that this dynamic endocytic network may also fulfil many of the functions of the Golgi TGN and that the latter may be absent in these stages. Although plausible, this comment needs further experimental support. For example, have the authors attempted to localize canonical makers of the TGN (e.g. GRIP proteins) in *T. brucei* BF and/or shown that exocytic carriers bud directly from the endosomes?

We agree with the criticism and have shortened the discussion accordingly and clearly marked it as speculation. However, we do not want to completely abandon our hypothesis.

The paragraph now reads:

Lines 740 – 751:

“Interestingly, we did not find any structural evidence of vesicular retrograde transport to the Golgi. Instead, the endosomal ‘highways’ extended throughout the posterior volume of the trypanosomes approaching the trans-Golgi interface. It is highly plausible that this region represents the convergence point where endocytic and biosynthetic membrane trafficking pathways merge. A comparable merging of endocytic and biosynthetic functions has been described for the TGN in plants. Different marker proteins for early and recycling endosomes were shown to be associated and/ or partially colocalized with the TGN suggesting its function in both secretory and endocytic pathways (reviewed in Minamino and Ueda, 2019). As we could not find structural evidence for the existence of a TGN we tentatively propose that trypanosomes may have shifted the central orchestrating function of the TGN as a sorting hub at the crossroads of biosynthetic and recycling pathways to the endosome. Although this is a speculative scenario, it is experimentally testable.”

Furthermore, we removed the lines 51 - 52, which included the suggestion of the TGN as a master regulator, from the abstract.

**Reviewer #2 (Public Review):**
The authors suggest that the African trypanosome endomembrane system has unusual organisation, in that the entire system is a single reticulated structure. It is not clear if this is thought to extend to the lysosome or MVB. There is also a suggestion that this unusual morphology serves as a trans-(post)Golgi network rather than the more canonical arrangement.The work is based around very high-quality light and electron microscopy, as well as utilising several marker proteins, Rab5A, 11 and 7. These are deemed as markers for early endosomes, recycling endosomes and late or pre-lysosomes. The images are mostly of high quality but some inconsistencies in the interpretation, appearance of structures and some rather sweeping assumptions make this less easy to accept. Two perhaps major issues are claims to label the entire endosomal apparatus with a single marker protein, which is hard to accept as certainly this reviewer does not really even know where the limits to the endosomal network reside and where these interface with other structures. There are several additional compartments that have been defined by Rob proteins as well, and which are not even mentioned. Overall I am unconvinced that the authors have demonstrated the main things they claim.The endomembrane system in bloodstream form T. brucei is clearly delimited. Compared to mammalian cells it is tidy and confined to the posterior part of the spindleshaped cell. The endoplasmic reticulum is linked to one side of the longitudinal cell axis, marked by the attached flagellum, while the mitochondrion locates to the opposite side. Glycosomes are easily identifiable as spheres, as are acidocalcisomes, which are smaller than glycosomes and – in electron micrographs – are characterized by high electron density. All these organelles extend beyond the nucleus, which is not the case for the endosomal compartment, the lysosome and the Golgi. The vesicles found in the posterior half of the trypanosome cell are quantitatively identifiable as COP1, CCVI or CCVII vesicles, or exocytic carriers. The lysosome has a higher degree of morphological plasticity, but this is not topic of the present work. Thus, the endomembrane system in *T. brucei* is comparatively well structured and delimited, which is why we have chosen trypanosomes as cell biological model.

We have published EP1::GFP as marker for the endosome system and flagellar pocket back in 2004. We have defined the fluid phase volume of the trypanosome endosome in papers published between 2002 and 2007. This work was not intended to represent the entirety of RAB proteins. We were only interested in 3 canonical markers for endosome subtypes. We do not claim anything that is not experimentally tested, we have clearly labelled our hypotheses as such, and we do not make sweeping assumptions.

The approaches taken are state-of-the-art but not novel, and because of the difficulty in fully addressing the central tenet, I am not sure how much of an impact this will have beyond the trypanosome field. For certain this is limited to workers in the direct area and is not a generalisable finding.

To the best of our knowledge, there is no published research that has employed 3D Tokuyasu or expansion microscopy (ExM) to label endosomes. The key takeaway from our study, which is the concept that "endosomes are continuous in trypanosomes" certainly is novel. We are not aware of any other report that has demonstrated this aspect.

The doubts formulated by the reviewer regarding the impact of our work beyond the field of trypanosomes are not timely. Indeed, our results, and those of others, show that the conclusions drawn from work with just a few model organisms is not generalisable. We are finally on the verge of a new cell biology that considers the plethora of evolutionary solutions beyond ophistokonts. We believe that this message should be widely acknowledged and considered. And we are certainly not the only ones who are convinced that the term "general relevance" is unscientific and should no longer be used in biology.

**Reviewer #3 (Public Review):**
Summary:As clearly highlighted by the authors, a key plank in the ability of trypanosomes to evade the mammalian host’s immune system is its high rate of endocytosis. This rapid turnover of its surface enables the trypanosome to ‘clean’ its surface removing antibodies and other immune effectors that are subsequently degraded. The high rate of endocytosis is likely reflected in the organisati’n and layout of the endosomal system in these parasites. Here, Link et al., sought to address this question using a range of light and three-dimensional electron microscopy approaches to define the endosomal organisation in this parasite.Before this study, the vast majority of our information about the make-up of the trypanosome endosomal system was from thin-section electron microscopy and immunofluorescence studies, which did not provide the necessary resolution and 3D information to address this issue. Therefore, it was not known how the different structures observed by EM were related. Link et al., have taken advantage of the advances in technology and used an impressive combination of approaches at the LM and EM level to study the endosomal system in these parasites. This innovative combination has now shown the interconnected-ness of this network and demonstrated that there are no ‘classical’ compartments within the endosomal system, with instead different regions of the network enriched in different protein markers (Rab5a, Rab7, Rab11).Strengths:This is a generally well-written and clear manuscript, with the data well-presented supporting the majority of the conclusions of the authors. The authors use an impressive range of approaches to address the organisation of the endosomal system and the development of these methods for use in trypanosomes will be of use to the wider parasitology community.I appreciate their inclusion of how they used a range of different light microscopy approaches even though for instance the dSTORM approach did not turn out to be as effective as hoped. The authors have clearly demonstrated that trypanosomes have a large interconnected endosomal network, without defined compartments and instead show enrichment for specific Rabs within this network.Weaknesses:My concerns are:i) There is no evidence for functional compartmentalisation. The classical markers of different endosomal compartments do not fully overlap but there is no evidence to show a region enriched in one or other of these proteins has that specific function. The authors should temper their conclusions about this point.

The reviewer is right in stating that Rab-presence does not necessarily mean Rabfunction. However, this assumption is as old as the Rab literature. That is why we have focused on the 3 most prominent endosomal marker proteins. We report that for endosome function you do not necessarily need separate membrane compartments. This is backed by our experiments.

ii) The quality of the electron microscopy work is very high but there is a general lack of numbers. For example, how many tomograms were examined? How often were fenestrated sheets seen? Can the authors provide more information about how frequent these observations were?

The fenestrated sheets can be seen in the majority of the 37 tomograms recorded of the posterior volume of the parasites. Furthermore, we have randomly generated several hundred tiled ( = very large) electron micrographs of bloodstream form trypanosomes for unbiased analyses of endomembranes. In these 2D-datasets the “footprint” of the fenestrated flat and circular cisternae is frequently detectable in the posterior cell area.

We now have included the corresponding numbers in all EM figure legends.

iii) The EM work always focussed on cells which had been processed before fixing. Now, I understand this was important to enable tracers to be used. However, given the dynamic nature of the system these processing steps and feeding experiments may have affected the endosomal organisation. Given their knowledge of the system now, the authors should fix some cells directly in culture to observe whether the organisation of the endosome aligns with their conclusions here.

This is a valid criticism; however, it is the cell culture that provides an artificial environment. As for a possible effect of cell harvesting by centrifugation on the integrity and functionality of the endosome system, we consider this very unlikely for one simple reason. The mechanical forces acting in and on the parasites as they circulate in the extremely crowded and confined environment of the mammalian bloodstream are obviously much higher than the centrifugal forces involved in cell preparation. This becomes particularly clear when one considers that the mass of the particle to be centrifuged determines the actual force exerted by the g-forces. Nevertheless, the proposed experiment is a good control, although much more complex than proposed, since tomography is a challenging technique. We have performed the suggested experiment and acquired tomograms of unprocessed cells. The corresponding data is now included as supplementary movie 2, 3 and 4. We refer to it in lines 202 – 206: To investigate potential impacts of processing steps (cargo uptake, centrifugation, washing) on endosomal organization, we directly fixed cells in the cell culture flask, embedded them in Epon, and conducted tomography. The resulting tomograms revealed endosomal organization consistent with that observed in cells fixed after processing (see Supplementary movie 2, 3, and 4).

We furthermore thank the reviewer for the experiment suggestion in the acknowledgments.

iv) The discussion needs to be revamped. At the moment it is just another run through of the results and does not take an overview of the results presenting an integrated view. Moreover, it contains reference to data that was not presented in the results.

We have improved the discussion accordingly.

**Recommendations for the authors:**
The reviewers concurred about the high calibre of the work and the importance of the findings.They raised some issues and made some suggestions to improve the paper without additional experiments - key issues include(1) Better referencing of the trypanosome endocytosis/ lysosomal trafficking literature.

The literature, especially the experimental and quantitative work, is very limited. We now provide a more complete set of references. However, we would like to mention that we had cited a recent review that critically references the trypanosome literature with emphasis on the extensive work done with mammalian cells and yeast.

(2) Moving the dSTORM data that detracts from otherwise strong data in a supplementary figure.

We have done this.

(3) Removal of the conclusion that the continuous endosome fulfils the functions of TGN, without further evidence.

As stated above, this was not a conclusion in our paper, but rather a speculation, which we have now more clearly marked as such. Lines 740 to 751 now read:

“Interestingly, we did not find any structural evidence of vesicular retrograde transport to the Golgi. Instead, the endosomal ‘highways’ extended throughout the posterior volume of the trypanosomes approaching the trans-Golgi interface. It is highly plausible that this region represents the convergence point where endocytic and biosynthetic membrane trafficking pathways merge. A comparable merging of endocytic and biosynthetic functions was already described for the TGN in plants. Different marker proteins for early and recycling endosomes were shown to be associated and/ or partially colocalized with the TGN suggesting its function in both secretory and endocytic pathways (reviewed in Minamino and Ueda, 2019). As we could not find structural evidence for the existence of a TGN we tentatively propose that trypanosomes may have shifted the central orchestrating function of the TGN as a sorting hub at the crossroads of biosynthetic and recycling pathways to the endosome. Although this is a speculative scenario, it is experimentally testable.”

(4) Broader discussion linking their findings to other examples of organelle maturation in eukaryotes (e.g cisternal maturation of the Golgi)

We have improved the discussion accordingly.

**Reviewer #1 (Recommendations For The Authors):**
What are the multi-vesicular vesicles that surround the marked endosomal compartments in Fig 1. Do they become labelled with fluid phase markers with longer incubations (e.g late endosome/ lysosomal)?

The function of MVBs in trypanosomes is still far from being clear. They are filled with fluid phase cargo, especially ferritin, but are devoid of VSG. Hence it is likely that MVBs are part of the lysosomal compartment. In fact, this part of the endomembrane system is highly dynamic. MVBs can be physically connected to the lysosome or can form elongated structures. The surprising dynamics of the trypanosome lysosome will be published elsewhere.

Figure 2. The compartments labelled with EP1::Halo are very poorly defined due to the low levels of expression of the reporter protein and/or sensitivity of detection of the Halo tag. Based on these images, it would be hard to conclude whether the endosome network is continuous or not. In this respect, it is unclear why the authors didn't use EP1-GFP for these analyses? Given the other data that provides more compelling evidence for a single continuous compartment, I would suggest removing Fig 2A.

We have used EP1::GFP to label the entire endosome system (Engstler and Boshart, 2004). Unfortunately, GFP is not suited for dSTORM imaging. By creating the EP1::Halo cell line, we were able to utilize the most prominent dSTORM fluorescent dye, Alexa 647. This was not primarily done to generate super resolution images, but rather to measure the dynamics of the GPI-anchored, luminal protein EP with single molecule precision. The results from this study will be published separately. But we agree with the reviewer and have relocated the dSTORM data to the supplementary material.

The observation that Rab5a/7 can be detected in the lumen of lysosome is interesting. Mechanistically, this presumably occurs by invagination of the limiting membrane of the lysosome. Is there any evidence that similar invagination of cytoplasmic markers occurs throughout or in subdomains of the endocytic network (possibly indicative of a 'late endosome' domain)?

So far, we have not observed this. The structure of the lysosome and the membrane influx from the endosome are currently being investigated.

The authors note that continuity of functionally distinct membrane compartments in the secretory/endocytic pathways has been reported in other protists (e.g T. cruzi). A particular example that could be noted is the endo-lysosomal system of *Dictyostelium* discoideum which mediates the continuous degradation and eventual expulsion of undigested material.

We tried to include this in the discussion but ultimately decided against it because the *Dictyostelium* system cannot be easily compared to the trypanosome endosome.

**Reviewer #2 (Recommendations For The Authors):**
AbstractNot sure that 'common' is the correct term here. Frequent, near-universal..... it would be true that endocytosis is common across most eukaryotes.

We have changed the sentence to “common process observed in most eukaryotes” (line 33).

Immune evasion - the parasite does not escape the immune system, but does successfully avoid its impact, at least at the population level.

We have replaced the word “escape” with “evasion” (line 35).

The third sentence needs to follow on correctly from the second. Also, more than Igs are internalised and potentially part of immune evasion, such as C3, Factor H, ApoL1 etcetera.

We believe that there may be a misunderstanding here. The process of endocytic uptake and lysosomal degradation has so far only been demonstrated in the context of VSGbound antibodies, which is why we only refer to this. Of course, the immune system comprises a wide range of proteins and effector molecules, all of which could be involved in immune evasion.

I do not follow the logic that the high flux through the endocytic system in trypanosomes precludes distinct compartmentalisation - one could imagine a system where a lot of steps become optimised for example. This idea needs expanding on if it is correct.

Membrane transport by vesicle transfer between several separate membrane compartments would be slower than the measured rate of membrane flux.

Again I am not sure 'efficient' on line 40. It is fast, but how do you measure efficiency? Speed and efficiency are not the same thing.

We have replaced the word “efficient” with “fast” (line 42).

The basis for suggesting endosomes as a TGN is unclear. Given that there are AP complexes, retromer, exocyst and other factors that are part of the TGN or at least post-G differentiation of pathways in canonical systems, this seems a step too far. There really is no evidence in the rest of the MS that seems to support this.

Yes, we agree and have clarified the discussion accordingly. We have not completely removed the discussion on the TGN but have labelled it more clearly as speculation.

I am aware I am being pedantic here, but overall the abstract seems to provide an impression of greater novelty than may be the case and makes several very bold claims that I cannot see as fully valid.

We are not aware of any claim in the summary that we have not substantiated with experiments, or any hypothesis that we have not explained.

Moreover, the concept of fused or multifunctional endosomes (or even other endomembrane compartments) is old, and has been demonstrated in metazoan cells and yeast. The concept of rigid (in terms of composition) compartments really has been rejected by most folks with maturation, recycling and domain structures already well-established models and concepts.

We agree that the (transient) presence of multiple Rab proteins decorating endosomes has been demonstrated in various cell types. This finding formed the basis for the endosomal maturation model in mammals and yeast, which has replaced the previous rigid compartment model.

However, we do not appreciate attempts to question the originality of our study by claiming that similar observations have been made in metazoans or yeast. This is simply wrong. There are no reports of a functionally structured, continuous, single and large endosome in any other system. The only membrane system that might be similar was described in the American parasite Trypanosoma cruzi, however, without the use of endosome markers or any functional analysis. We refer to this study in the discussion.

In summary, the maturation model falls short in explaining the intricacies of the membrane system we have uncovered in trypanosomes. Therefore, one plausible interpretation of our data is that the overall architecture of the trypanosome endosomes represents an adaptation that enables the remarkable speed of plasma membrane recycling observed in these parasites. In our view, both our findings and their interpretation are novel and worth reporting. Again, modern cell biology should recognize that evolution has developed many solutions for similar processes in cells, about whose diversity we have learned almost nothing because of our reductionist view. A remarkable example of this are the Picozoa, tiny bipartite eukaryotes that pack the entire nutritional apparatus into one pouch and the main organelles with the locomotor system into the other. Another one is the “extreme” cell biology of many protozoan parasites such as Giardia, Toxpoplasma or Trypanosoma.

Higher plants have been well characterised, especially at the level of Rab/Arf proteins and adaptins.

We now mention plant endosomes in our brief discussion of the trypanosome TGN. Lines 744 – 747:

“A comparable merging of endocytic and biosynthetic functions was already described for the TGN in plants. Different marker proteins for early and recycling endosomes were shown to be associated and/ or partially colocalized with the TGN suggesting its function in both secretory and endocytic pathways (reviewed in Minamino and Ueda, 2019).”

The level of self-citing in the introduction is irritating and unscholarly. I have no qualms with crediting the authors with their own excellent contributions, but work from Dacks, Bangs, Field and others seems to be selectively ignored, with an awkward use of the authors' own publications. Diversity between organisms for example has been a mainstay of the Dacks lab output, Rab proteins and others from Field and work on exocytosis and late endosomal systems from Bangs. These efforts and contributions surely deserve some recognition?

This is an original article and not a review. For a comprehensive overview the reviewer might read our recent overview article on exo- and endocytic pathways in trypanosomes, in which we have extensively cited the work of Mark Field, Jay Bangs and Joel Dacks. In the present manuscript, we have cited all papers that touch on our results or are otherwise important for a thorough understanding of our hypotheses. We do not believe that this approach is unscientific, but rather improves the readability of the manuscript. Nevertheless, we have now cited additional work.

For the uninitiated, the posterior/anterior axis of the trypanosome cell as well as any other specific features should be defined.

In lines 102 - 110 we wrote:

“This process of antibody clearance is driven by hydrodynamic drag forces resulting from the continuous directional movement of trypanosomes (Engstler et al., 2007). The VSG-antibody complexes on the cell surface are dragged against the swimming direction of the parasite and accumulate at the posterior pole of the cell. This region harbours an invagination in the plasma membrane known as the flagellar pocket (FP) (Gull, 2003; Overath et al., 1997). The FP, which marks the origin of the single attached flagellum, is the exclusive site for endo- and exocytosis in trypanosomes (Gull, 2003; Overath et al., 1997). Consequently, the accumulation of VSG-antibody complexes occurs precisely in the area of bulk membrane uptake.”

We think this sufficiently introduces the cell body axes.

I don't understand the comment concerning microtubule association. In mammalian cells, such association is well established, but compartments still do not display precise positioning. This likely then has nothing to do with the microtubule association differences.

We have clarified this in the text (lines 192 – 199). There is no report of cytoplasmic microtubules in trypanosomes. All microtubules appear to be either subpellicular or within the flagellum. To maintain the structure and position of the endosomal apparatus, they should be associated either with subpellicular microtubules, as is the case with the endoplasmic reticulum, or with the more enigmatic actomyosin system of the parasites.We have been working on the latter possibility and intend to publish a follow-up paper to the present manuscript.

The inability to move past the nucleus is a poor explanation. These compartments are dynamic. Even the nucleus does interesting things in trypanosomes and squeezes past structures during development in the tsetse fly.

The distance between the nucleus and the microtubule cytoskeleton remains relatively constant even in parasites that squeeze through microfluidic channels. This is not unexpected as the nucleus can be highly deformed. A structure the size of the endosome will not be able to physically pass behind the nucleus without losing its integrity. In fact, the recycling apparatus is never found in the anterior part of the trypanosome, most probably because the flagellar pocket is located at the posterior cell pole.

L253 What is the evidence that EP1 labels the entire FP and endosomes? This may be extensive, but this claim requires rather more evidence. This is again suggested at l263. Again, please forgive me for being pedantic, but this is an overstatement unless supported by evidence that would be incredibly difficult to obtain. This is even sort of acknowledged on l271 in the context of non-uniform labelling. This comes again in l336.

The evidence that EP1 labels the entire FP and endosomes is presented here: Engstler and Boshart, 2004; 10.1101/gad.323404.

Perhaps I should refrain from comments on the dangers of expansion microscopy, or asking what has actually been gained here. Oddly, the conclusion on l290 is a fair statement that I am happy with.

An in-depth discussion regarding the advantages and disadvantages of expansion microscopy is beyond the manuscript's intended scope. Our approach involved utilizing various imaging techniques to confirm the validity of our findings. We appreciate that our concluding sentence is pleasing.

F2 - The data in panel A seem quite poor to me. I also do not really understand why the DAPI stain in the first and second columns fails to coincide or why the kinetoplast is so diffuse in the second row. The labelling for EP1 presents as very small puncta, and hence is not evidence for a continuum. What is the arrow in A IV top? The data in panel B are certainly more in line with prior art, albeit that there is considerable heterogeneity in the labelling and of the FP for example. Again, I cannot really see this as evidence for continuity. There are gaps.... Albeit I accept that labelling of such structures is unlikely to ever be homogenous.

We agree that the dSTORM data represents the least robust aspect of the findings we have presented, and we concur with relocating it to the supplementary material.

F3 - Rather apparent, and specifically for Rab7, that there is differential representation - for example, Cell 4 presents a single Rab7 structure while the remaining examples demonstrate more extensive labelling. Again, I am content that these are highly dynamic strictures but this needs to be addressed at some level and commented upon. If the claim is for continuity, the dynamics observed here suggest the usual; some level of obvious overlap of organellar markers, but the representation in F3 is clever but not sure what I am looking at. Moreover, the title of the figure is nothing new. What is also a bit odd is that the extent of the Rab7 signal, and to some extent the other two Rabs used, is rather variable, which makes this unclear to me as to what is being detected. Given that the Rab proteins may be defining microdomains or regions, I would also expect a region of unique straining as well as the common areas. This needs to at least be discussed.

The differences in the representation result from the dynamics of the labelled structures. Therefore, we have selected different cells to provide examples of what the labelling can look like. We now mention this in the results section.

The overlap of the different Rab signals was perhaps to be expected, but we now have demonstrated it experimentally. Importantly, we performed a rigorous quantification by calculating the volume overlaps and the Pearson correlation coefficients.

In previous studies the data were presented as maximal intensity projections, which inherently lack the complete 3D information.

We found that Rab proteins define microdomains and that there are regions of unique staining as well as common areas, as shown in Figure 3. The volumes do not completely overlap. This is now more clearly stated in lines 315 – 319:

“These objects showed areas of unique staining as well as partially overlapping regions. The pairwise colocalization of different endosomal markers is shown in Figure 3 A, XI - XIII and 3 B. The different cells in Figure 3 B were selected to represent the dynamic nature of the labelled structures. Consequently, the selected cells provide a variety of examples of how the labelling can appear.”

This had already been stated in lines 331 – 336:

“In summary, the quantitative colocalization analyses revealed that on the one hand, the endosomal system features a high degree of connectivity, with considerable overlap of endosomal marker regions, and on the other hand, TbRab5A, TbRab7, and TbRab11 also demarcate separated regions in that system. These results can be interpreted as evidence of a continuous endosomal membrane system harbouring functional subdomains, with a limited amount of potentially separated early, late or recycling endosomes.”

F4-6 - Fabulous images. But a couple of issues here; first, as the authors point out, there is distance between the gold and the antigen. So, this of course also works in the z-plane as well as the x/y-planes and some of the gold may well be associated with membraneous figures that are out of the plane, which would indicate an absence of colinearity on one specific membrane. Secondly, in several instances, we have Rab7 essentially mixed with Rab11 or Rab5 positive membrane. While data are data and should be accepted, this is difficult to reconcile when, at least to some level, Rab7 is a marker for a late-endosomal structure and where the presence of degradative activity could reside. As division of function is, I assume, the major reason for intracellular compartmentalisation, such a level of admixture is hard to rationalise. A continuum is one thing but the data here seem to be suggesting something else, i.e. almost complete admixture.

We are grateful for the positive feedback regarding the image quality. It is true that the "linkage error," representing the distance between the gold and the antigen, also functions to some extent in the z-axis. However, it's important to note that the zdimension of the section in these Figures is 55 nm. Nevertheless, it's interesting to observe that membranes, which may not be visible within the section itself but likely the corresponding Rab antigen, is discernible in Figure 4C (indicated by arrows).

We have clarified this in lines 397 – 400:

“Consequently, gold particles located further away may represent cytoplasmic TbRab proteins or, as the “linkage error” can also occur in the z-plane, correspond to membranes that are not visible within the 55 nm thickness of the cryosection (Figure 4, panel C, arrows). “

The coexistence of different Rabs is most likely concentrated in regions where transitions between different functions are likely. Our focus was primarily on imaging membranes labelled with two markers. We wanted to show that the prevailing model of separate compartments in the trypanosome literature is not correct.

F7 - Not sure what this adds beyond what was published by Grunfelder.

First, this figure is an important control that links our results to published work (Grünfelder et al. (2003)). Second, we include double staining of cargo with Rab5, Rab7, and Rab11, whereas Grünfelder focused only on Rab11. Therefore, our data is original and of such high quality that it warrants a main figure.

F8 - and l583. This is odd as the claim is 'proof' which in science is a hard thing to claim (and this is definitely not at a six sigma level of certainty, as used by the physics community). However, I am seeing structures in the tomograms which are not contiguous - there are gaps here between the individual features (Green in the figure).

We have replaced the term "proof". It is important to note that the structures in individual tomograms cannot all be completely continuous because the sections are limited to a thickness of 250 nm. Therefore, it is likely that they have more connectivity above and below the imaged section. Nevertheless, we believe that the quality of the tomograms is satisfactory, considering that 3D Tokuyasu is a very demanding technique and the production of serial Tokuyasu tomograms is not feasible in practice.

Discussion - Too long and the self-citing of four papers from the corresponding author to the exclusion of much prior work is again noted, with concerns about this as described above. Moreover, at least four additional Rab proteins are known associated with the trypanosome endosomal system, 4, 5B, 21 and 28. These have been completely ignored.

We have outlined our position on referencing in original articles above. We also explained why we focused on the key marker proteins associated with early (Rab5), late (Rab7) and recycling endosomes (Rab11). We did not ignore the other Rabs, we just did not include them in the present study.

Overall this is disappointing. I had expected a more robust analysis, with a clearer discussion and placement in context. I am not fully convinced that what we have here is as extreme as claimed, or that we have a substantial advance. There is nothing here that is mechanistic or the identification of a new set of gene products, process or function.

We do not think that this is constructive feedback.

This MS suggests that the endosomal system of African trypanosomes is a continuum of membrane structures rather than representing a set of distinct compartments. A combination of light and electron microscopy methods are used in support. The basic contention is very challenging to prove, and I'm not convinced that this has been. Furthermore, I am also unclear as to the significance of such an organisation; this seems not really addressed.

We acknowledge and respect varying viewpoints, but we hold a differing perspective in this matter. We are convinced that the data decisively supports our interpretation. May future work support or refute our hypothesis.

**Reviewer #3 (Recommendations For The Authors):**
Line 81 - delete 's

Done.

Generally, the introduction was very well written and clearly summarised our current understanding but the paragraph beginning line 134 felt out of place and repeated some of the work mentioned earlier.

We have removed this paragraph.

For the EM analysis throughout quantification would be useful as highlighted in the public review. How many tomograms were examined, and how often were types of structures seen? I understand the sample size is often small but this would help the reader appreciate the diversity of structures seen.

We have included the numbers.

Following on from this how were the cells chosen for tomogram analysis? For example, the dividing cell in 1D has palisades associating with the new pocket - is this commonly seen? Does this reflect something happening in dividing cells. This point about endosomal division was picked up in the discussion but there was little about in the main results.

This issue is undoubtedly inherent to the method itself, and we have made efforts to mitigate it by generating a series of tomograms recorded randomly. We have refrained from delving deeper into the intricacies of the cell cycle in this manuscript, as we believe that it warrants a separate paper.

As the authors prosecute, the co-localisation analysis highlights the variable nature of the endosome and the overlap of different markers. When looking at the LM analysis, I was struck by the variability in the size and number of labelled structures in the different cells. For example, in 3A Rab7 is 2 blobs but in 3B Cell 1 it is 4/5 blobs. Is this just a reflection of the increase in the endosome during the cell cycle?

The variability in representation is a direct consequence of the dynamic nature of the labelled structures. For this reason, we deliberately selected different cells to represent examples of how the labelling can look like. We have decided not to mention the dynamics of the endosome during the cell cycle. This will be the subject of a further report.

Moreover, Rab 11 looks to be the marker covering the greatest volume of the endosomal system - is this true? I think there's more analysis of this data that could be done to try and get more information about the relative volumes etc of the different markers that haven't been drawn out. The focus here is on the co-localisation.

Precisely because we recognize the importance of this point, we intend to turn our attention to the cell cycle in a separate publication.

I appreciate that it is an awful lot of work to perform the immuno-EM and the data is of good quality but in the text, there could be a greater effort to tie this to the LM data. For example, from the Rab11 staining in LM you would expect this marker to be the most extensive across the networks - is this reflected in the EM?For the immuno-EM there were no numbers, the authors had measured the position of the gold but what was the proportion of gold that was in/near membranes for each marker? This would help the reader understand both the number of particles seen and the enrichment of the different regions.

Our original intent was to perform a thorough quantification (using stereology) of the immuno-EM data. However, we later realized that the necessary random imaging approach is not suitable for Tokuyasu sections of trypanosomes. In short, the cells are too far apart, and the cell sections are only occasionally cut so that the endosomal membranes are sufficiently visible. Nevertheless, we continue to strive to generate more quantitative data using conventional immuno-EM.

The innovative combination of Tokuyasu tomograms with immuno-EM was great. I noted though that there was a lack of fenestration in these models. Does this reflect the angle of the model or the processing of these samples?

We are grateful to the referee, as we have asked ourselves the same question. However, we do not attribute the apparent lack of fenestration to the viewing angle, since we did not find fenestration in any of the Tokuyasu tomograms. Our suspicion is more directed towards a methodological problem. In the Tokuyasu workflow, all structures are mainly fixed with aldehydes. As a result, lipids are only effectively fixed through their association with membrane proteins. We suggest that the fenestration may not be visible because the corresponding lipids may have been lost due to incomplete fixation.

We now clearly state this in the lines 563 – 568.

“Interestingly, these tomograms did not exhibit the fenestration pattern identified in conventional electron tomography. We suspect that this is due to methodological reasons. The Tokuyasu procedure uses only aldehydes to fix all structures. Consequently, effective fixation of lipids occurs only through their association with membrane proteins. Thus, the lack of visible fenestration is likely due to possible loss of lipids during incomplete fixation.”

The discussion needs to be reworked. Throughout it contains references to results not in the main results section such as supplementary movie 2 (line 735). The explicit references to the data and figures felt odd and more suited to the results rather than the discussion. Currently, each result is discussed individually in turn and more effort needs to be made to integrate the results from this analysis here but also with previous work and the data from other organisms, which at the moment sits in a standalone section at the end of the discussion.

We have improved the discussion and removed the previous supplementary movies 2 and 3. Supplementary movie 1 is now mentioned in the results section.

Line 693 - There was an interesting point about dividing cells describing the maintenance of endosomes next to the old pocket. Does that mean there was no endosome by the new pocket and if so where is this data in the manuscript? This point relates back to my question about how cells were chosen for analysis - how many dividing cells were examined by tomography?

The fate of endosomes during the cell cycle is not the subject of this paper. In this manuscript we only show only one dividing cell using tomography. An in-depth analysis focusing on what happens during the cell cycle will be published separately.

Line 729 - I'm unclear how this represents a polarization of function in the flagellar pocket. The pocket I presume is included within the endosomal system for this analysis but there was no specific mention of it in the results and no marker of each position to help define any specialisation. From the results, I thought the focus was on endosomal co-localisation of the different markers. If the authors are thinking about specialisation of the pocket this paper from Mark Field shows there is evidence for the exocyst to be distributed over the entire surface of the pocket, which is relevant to the discussion here. Boehm, C.M. et al. (2017) The trypanosome exocyst: a conserved structure revealing a new role in endocytosis. PLoS Pathog. 13, e1006063

We have formulated our statement more cautiously. However, we are convinced that membrane exchange cannot physically work without functional polarization of the pocket. We know that Rab11, for example, is not evenly distributed on the pocket. By the way, in Boehm et al. (2017) the exocyst is not shown to cover the entire pocket (as shown in Supplementary Video 1).

We now refer to Boehm et al. (Lines 700 – 703):

“Boehm et al (2017) report that in the flagellar pocket endocytic and exocytic sites are in close proximity but do not overlap. We further suggest that the fusion of EXCs with the flagellar pocket membrane and clathrin-mediated endocytosis take place on different sites of the pocket. This disparity explains the lower colocalization betweenTbRab11 and TbRab5A.”

Line 735 - link to data not previously mentioned I think. When I looked at this data I couldn't find a key to explain what all the different colours related to.

We have removed the previous supplementary movies 2 and 3. We now reference supplementary movie 1 in the results section.